# Calcium signaling through a transient receptor channel is important for *Toxoplasma gondii* growth

**Karla Marie Márquez-Nogueras[1,2], Miryam Andrea Hortua Triana[1], Nathan M Chasen[1], Ivana Y Kuo[2], Silvia NJ Moreno[1,3]\***

[1]Center for Tropical and Emerging Global Diseases, University of Georgia, Athens, United States; [2]Department of Cell and Molecular Physiology, Stritch School of Medicine, Loyola University Chicago, Maywood, United States; [3]Department of Cellular Biology, University of Georgia, Athens, United States

**Abstract** Transient receptor potential (TRP) channels participate in calcium ion ($Ca^{2+}$) influx and intracellular $Ca^{2+}$ release. TRP channels have not been studied in *Toxoplasma gondii* or any other apicomplexan parasite. In this work, we characterize TgGT1_310560, a protein predicted to possess a TRP domain (TgTRPPL-2), and determined its role in $Ca^{2+}$ signaling in *T. gondii*, the causative agent of toxoplasmosis. TgTRPPL-2 localizes to the plasma membrane and the endoplasmic reticulum (ER) of *T. gondii*. The $\Delta TgTRPPL-2$ mutant was defective in growth and cytosolic $Ca^{2+}$ influx from both extracellular and intracellular sources. Heterologous expression of TgTRPPL-2 in HEK-3KO cells allowed its functional characterization. Patching of ER-nuclear membranes demonstrates that TgTRPPL-2 is a non-selective cation channel that conducts $Ca^{2+}$. Pharmacological blockers of TgTRPPL-2 inhibit $Ca^{2+}$ influx and parasite growth. This is the first report of an apicomplexan ion channel that conducts $Ca^{2+}$ and may initiate a $Ca^{2+}$ signaling cascade that leads to the stimulation of motility, invasion, and egress. TgTRPPL-2 is a potential target for combating toxoplasmosis.

**\*For correspondence:**
smoreno@uga.edu

**Competing interests:** The authors declare that no competing interests exist.

## Introduction

$Ca^{2+}$ signaling is universal and forms part of the signaling pathways that activate or modulate a variety of physiological responses like gene transcription, muscle contraction, cell differentiation, and proliferation (*Berridge et al., 2003*). $Ca^{2+}$ signals can be generated through the opening of ion channels that allow the downward flow of $Ca^{2+}$ from either outside the cell or from intracellular stores like the endoplasmic reticulum (ER; *Clapham, 2007*).

*Toxoplasma gondii* is an intracellular parasite from the Apicomplexa phylum that causes toxoplasmosis in humans (*Blader et al., 2015*). Infection with *T. gondii* may lead to severe complications in immunocompromised patients like encephalitis, myocarditis, and death (*Weiss and Dubey, 2009*). The *T. gondii* tachyzoite engages in a lytic cycle directly responsible for the pathogenicity of the infection as it results in lysis of host cells (*Black and Boothroyd, 2000*). The lytic cycle consists of active invasion of host cells, replication inside a parasitophorous vacuole (PV) and egress to search for a new host cell to invade. $Ca^{2+}$ signals resulting from $Ca^{2+}$ entry or from intracellular release trigger a signaling cascade in the parasite that culminates in the stimulation of essential features of its lytic cycle, like motility, invasion, egress, and secretion of proteins essential for attachment to the host cell (*Hortua Triana et al., 2018b*; *Lourido and Moreno, 2015*).

Previous work from our lab showed the presence of a $Ca^{2+}$ entry activity at the plasma membrane of *T. gondii* tachyzoites that was functional in extracellular (*Pace et al., 2014*) and intracellular replicating tachyzoites (*Vella et al., 2021*). The application of voltage-operated $Ca^{2+}$ channel blockers

such as nifedipine inhibited ~80% of $Ca^{2+}$ influx, and the residual $Ca^{2+}$ entry activity suggested the potential existence of more than one channel at the plasma membrane of *T. gondii* (*Pace et al., 2014*). The molecular nature of these channels has remained elusive. In addition, $Ca^{2+}$ efflux from the ER into the parasite's cytosol was revealed upon inhibition of the sarcoplasmic-endoplasmic reticulum $Ca^{2+}$ pump (SERCA) with thapsigargin (Thap) (*Moreno and Zhong, 1996*; *Pace et al., 2014*). This efflux activity has not been molecularly characterized.

Transient receptor potential (TRP) channels are a large family of ~33 cation-permeable channels grouped into seven subfamilies based on their gene sequence (*Nilius and Owsianik, 2011*). TRP channels can be activated by a multitude of stimuli and are involved in a wide range of cellular functions (*Zhou, 2009*). Most TRP channels are permeable to $Ca^{2+}$, and all of them are permeable to monovalent cations (*Zhou, 2009*). Some TRP channels can participate in $Ca^{2+}$ influx as well as $Ca^{2+}$ release from intracellular stores (*Koulen et al., 2002*; *Venkatachalam and Montell, 2007*). Mutations in these molecules are associated with a diverse set of diseases due to their wide distribution in various tissues and their roles in pathological conditions like cancer, making these channels important therapeutic targets (*Samanta et al., 2018*). The polycystin TRP (TRPP) subfamily of proteins are implicated in autosomal-dominant polycystic kidney disease (ADPKD) (*Wang et al., 2019*).

Predicted protein sequences with TRP domains have been found in most parasitic protozoa, although in lower numbers and types than in other organisms (*Wolstenholme et al., 2011*). This could be the result of the evolutionary distance between the species studied or because of loss of specific functions resulting from evolution of the parasitic lifestyle (*Wolstenholme et al., 2011*). A genome analysis of a number of pathogenic protozoan parasites (*Prole and Taylor, 2011*) searching for genes with homology to mammalian $Ca^{2+}$ channels identified two *T. gondii* hypothetical genes (TgGT1_247370 and TgGT1_310560) with homologous regions to the TRPP family (*Boucher and Sandford, 2004*). We termed these genes *TgTRPPL-1* and *TgTRPPL-2*. Previous work from our laboratory localized TgTRPPL-1 to the ER with high-resolution tags due to its low level of expression (*Hortua Triana et al., 2018a*).

In this work, we characterize TgTRPPL-2 in *T. gondii*, which represents the first TRP cation channel studied in any apicomplexan parasite. Using reverse genetic approaches, we determine the role of TgTRPPL-2 in the lytic cycle of the parasite. We also characterize the electrophysiological features of TgTRPPL-2 and its role in $Ca^{2+}$ transport and, interestingly, find that pharmacological agents that block the activity of TgTRPPL-2 also inhibit cytosolic $Ca^{2+}$ influx in the parasite and parasite growth. TgTRPPL-2 emerges as one of the molecular entities involved in initiating $Ca^{2+}$ signals in *T. gondii*.

## Results

### TgTRPPL-2 (TgGT1_310560) localizes to the plasma membrane and the endoplasmic reticulum

Two genes in the *T. gondii's* genome annotated as hypothetical proteins possess polycystic kidney disease (PKD) domains, which are characteristic of the Subfamily P (polycystin) of TRP channels. Mammalian TRPP channels contain six transmembrane domains with a large extracellular loop between the first and second transmembrane domain (*Montell, 2005*). We termed these proteins in *T. gondii* TgTRPPL-1 (TgGT1_247370) and TgTRPPL-2 (TgGT1_310560). Using BLASTp to compare the amino acid sequences of the human *PC2* and *TgTRPPL-2* showed low-sequence homology (21.7%), even within the PKD domains. The *TgGT1_310560* gene predicts the expression of a protein of 2191 amino acids with an apparent molecular weight of 237 kDa and 14 transmembrane domains. The predicted topology (*Omasits et al., 2014*) showed a large extracellular loop between the first and second transmembrane domains, which is characteristic of PKD channels (*Figure 1A, B*). Because our initial analysis showed low-sequence homology, we next analyzed the amino acid sequence using the software HHPred, which searches for homology based on protein sequence and secondary structure (*Söding, 2005*). Sequence analysis of TgTRPPL-2 showed high homology to human PC2, and the top 10 hits obtained were PC2 homologous from a variety of organisms (*Supplementary file 1*).

To investigate the localization of TgTRPPL-2, we introduced the high-affinity tag smHA (*Hortua Triana et al., 2018a*) at the 3′ terminus of the *TgTRPPL-2* locus and isolated TgTRPPL-2-smHA cell clones. Carboxy-terminus tagging was done in the parental line RHTatiΔku80 (*TatiΔku80*),

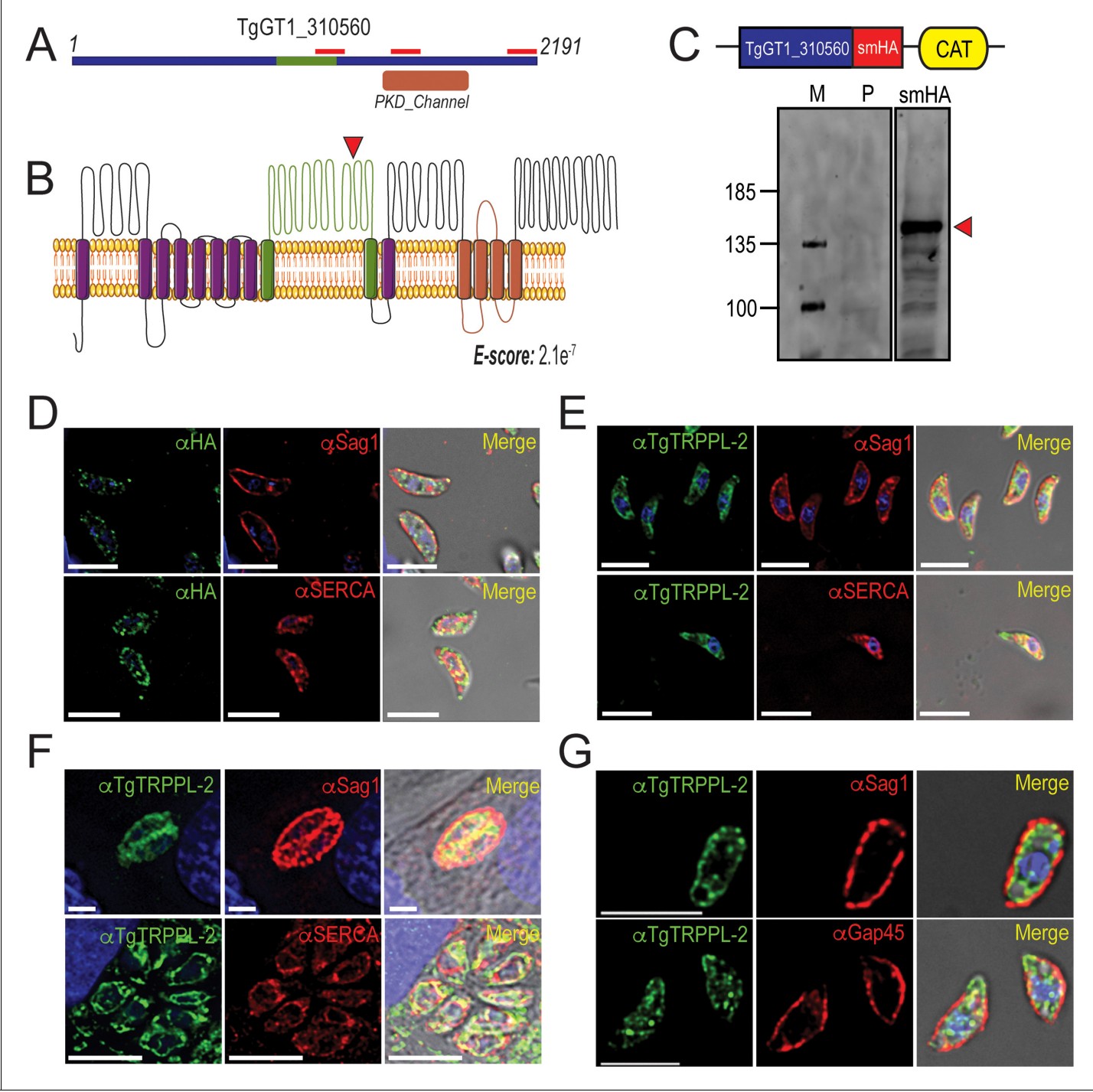

**Figure 1.** TgTRPPL-2 localizes to the endoplasmic reticulum and periphery of *T. gondii*. (**A**) Schematic representation of the InterPro Domain annotation of TgTRPPL-2 in the GT1 strain. Red line indicates coverage of the sequence by mass spectrometry. (**B**) Predicted topology for TgTRPPL-2 in GT1 strain. Model was generated with the Protter application (***Omasits et al., 2014***). The PKD domain is shown in orange. The domain used to generate antibodies is highlighted in green. Predicted Transmembrane Domains (TMDs) are highlighted in purple. The red arrowhead indicates the predicted cleavage site for TgTRPPL2. (**C**) Schematic representation of C-terminal tagging of TgTRPPL-2 in TatiΔKu80 parasites and western blots of TgTRPPL-2-smHA membranes using αHA (1:1000) showing a major band at approximately 150 kDa (*red arrowhead*). (**D**) Immunofluorescence analysis (IFA) of extracellular tachyzoites using αHA antibody and co-localization with αSAG1 (1:1000) and αSERCA (1:1000) showing partial co-localization with both markers. (**E**) IFAs of extracellular and intracellular (**F**) tachyzoites respectively with αTgTRPPL-2 (1:100) showing labeling of the protein at the periphery, co-localized with αSAG1 (1:1000) and with αTgSERCA (1:1000). (**G**) Super-resolution of extracellular IFAs using αTgTRPPL-2 (1:100) co-localized with αSAG1 (1:1000) but not with αGAP45 (1:1000). Scale bars in D-G represent 5 µm.

*Figure 1 continued on next page*

*Figure 1 continued*

The online version of this article includes the following source data and figure supplement(s) for figure 1:

**Source data 1.** Mass spectrometry results.
**Figure supplement 1.** Validation of C-terminal tagging of TgTRPPL-2-smHA.
**Figure supplement 1—source data 1.** Summary statistics of Pearson's coefficient analysis.

which favors homologous recombination (*Sheiner et al., 2011*). Correct incorporation of the tag in the TgTRPPL-2-smHA line was validated by PCR (*Figure 1—figure supplement 1A*) and western blot analysis using anti-HA antibodies (*Figure 1C*). A band of approximately ~150 kDa was observed in lysates of TgTRPPL-2-smHA tachyzoites, which is nearly 87 kDa smaller than the predicted size of 237 kDa without taking into account the smHA tag (~39 kDa).

To further demonstrate that the protein band observed in the western blot analysis corresponded to the tagged TgTRPPL-2 gene, we performed immunoprecipitations with anti-HA of lysates from the TgTRPPL-2-smHA cells. The immunoprecipitated samples were separated in a PAGE gel. The ~150 kDa band was excised and analyzed by mass spectrometry *Figure 1—figure supplement 1B red boxes* (*Supplementary file 2*). The peptides recovered are shown matching the TgGT1_310560-predicted protein sequence, which cover ~700 amino acids of the C-terminus domain (*Figure 1A*, *red bars*). This result indicated that the TgGT1_310560 is likely cleaved, a characteristic common with other TRP channels (*Merrick et al., 2014*). The predicted cleavage site is shown in *Figure 1B*.

We next investigated the cellular localization of TgTRPPL-2. Immunofluorescence analysis (IFA) of extracellular and intracellular parasites showed that TgTRPPL-2-smHA localizes to peripheral vesicles close to the plasma membrane and to the ER (*Figure 1D*). Some co-localization with the plasma membrane surface antigen (SAG1) and the sarcco-endoplasmic reticulum $Ca^{2+}$ ATPase (TgSERCA) (ER marker) was observed in IFAs (*Figure 1D*) while the vesicular pattern did now show co-localization with dense granules (*Figure 1—figure supplement 1D*). However, considering the low level of expression of TgTRPPL-2, it was difficult to draw definitive conclusions about its localization.

We next generated polyclonal antibodies against a fragment peptide of TgTRPPL-2, indicated in *Figure 1A* (*highlighted in green*). The peptide was expressed in bacteria, purified, and used for immunization of mice. Mouse serum was isolated and affinity purified prior to its use. The localization at the periphery of extracellular tachyzoites was further confirmed by co-localization with αSAG1 (*Figure 1E–G*). In addition, extracellular and intracellular tachyzoites showed staining that co-localized with TgSERCA (*Figure 1E, F*), supporting ER localization. IFAs of intracellular tachyzoites showed that TgTRPPL-2 co-localized with αSAG1 and αSERCA (*Figure 1F*). Super-resolution images with the anti-TgTRPPL-2 antibody showed localization to the periphery in close contact with the SAG1 marker (*Figure 1—figure supplement 1E*). TgTRPPL-2 did not appear to overlap with GAP45 (*Figure 1—figure supplement 1E*), which is supported by Pearson's coefficient quantification.

In summary, TgTRPPL-2 is expressed in *T. gondii* tachyzoites, is likely post-translationally cleaved, and localizes to the ER and the periphery.

## TgTRPPL-2 is important for growth, invasion, and egress of *T. gondii*

With the aim of investigating the physiological role of TgTRPPL-2 in *T. gondii,* we generated the ΔTgTRPPL-2 mutant using the CRISPR-Cas9 approach (*Shen et al., 2017*) to disrupt the transcription of *TgTRPPL-2* by inserting a dihydrofolate reductase-thymidylate synthase (DHFR) cassette in the *TgTRPPL-2* genomic locus (*Figure 2A*). Genetic controls for the insertion were done by PCR (*Figure 2—figure supplement 1A*) and qPCR, which showed a significant decrease in the levels of *TgTRPPL-2* transcripts (*Figure 2B*).

We next complemented the ΔTgTRPPL-2 mutant with Cosmid PSBLZ13 (*Vinayak et al., 2014*) that contains the whole genomic locus of the *TgTRPPL-2* gene and generated the cell line ΔTgTRPPL-2-trppl2. Controls for the expression of *TgTRPPL-2* were done by qPCR (*Figure 2B*) and IFAs, which further confirmed the identity of the tagged gene, as it was not expressed in the ΔTgTRPPL-2 mutants and was present in the complemented line ΔTgTRPPL-2-trppl2 (*Figure 2C*). Further validation of the absence of expression of TgTRPPL-2 and its complementation is shown in *Figure 2—figure supplement 1* with western blots analyses with mouse α-TgTRPPL-2 antibody and

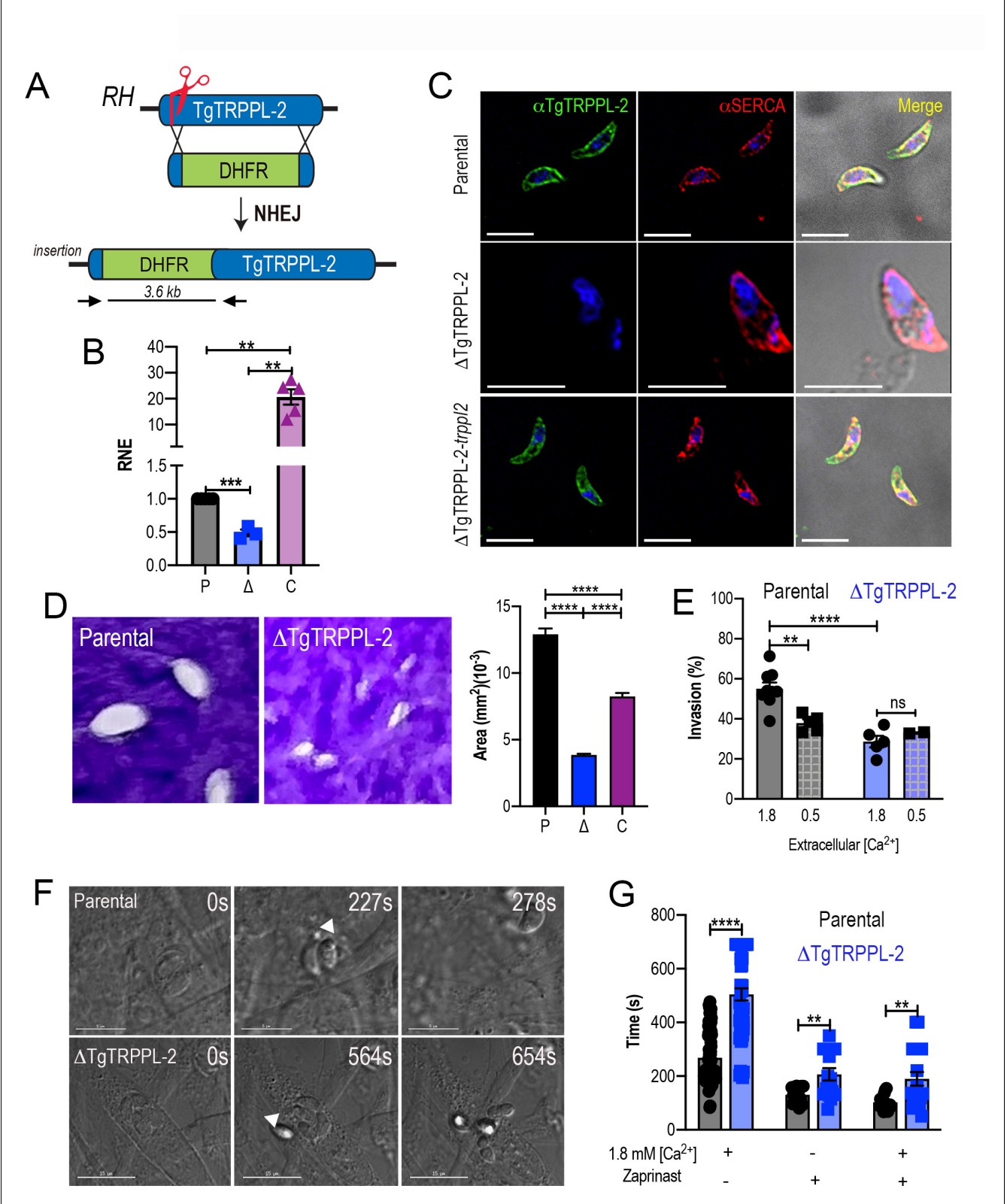

**Figure 2.** The role of TgTRPPL-2 in *T. gondii* growth. (**A**) Schematic representation of the generation of Δ*TgTRPPL-2* in the *T. gondii* RH strain. (**B**) qPCR of total RNA from Δ*TgTRPPL-2* (Δ), Δ*TgTRPPL-2-trppl2* (C), and parental strains (P) using primers upstream and downstream of the insertion site of the dihydrofolate reductase-thymidylate synthase cassette. (**C**) Immunofluorescence analysis of extracellular parasites showing plasma membrane labeling with αTgTRPPL-2 (1:1000) and co-localization with αSERCA (1:1000). (**D**) Plaque assays of parental (P), Δ*TgTRPPL-2* (Δ), and Δ*TgTRPPL-2-*

*Figure 2 continued on next page*

*Figure 2 continued*

*trppl2* (C) parasites. Quantification of plaque sizes from three independent biological experiments using Student's *t*-test. Values are means ± SEM. ****p<0.0001. (E) Red green assays of parental, and ΔTgTRPPL-2 cells quantifying invaded and attached intracellular parasites. Assays were done at two concentrations of extracellular $Ca^{2+}$: 0.5 and 1.8 mM. Values are means ± SEM. **p<0.001, ****p<0.0001. (F) Time to egress stimulated by saponin/$Ca^{2+}$ at 1.8 mM extracellular $Ca^{2+}$ of both parental and the ΔTgTRPPL-2 mutant. (G) Statistical analysis of average egress time stimulated by saponin or Zaprinast in the presence or absence of extracellular $Ca^{2+}$. Analysis was performed from three independent biological replicates using Student's *t*-test. Values are means ± SEM, **p<0.003, ****p<0.0001. *Black bars* represent parental strain, *blue bars* represent ΔTgTRPPL-2. Scale bars for C represent 5 μm.

The online version of this article includes the following source data and figure supplement(s) for figure 2:

**Source data 1.** Statistical analysis of data.
**Figure supplement 1.** Validation of the anti-TgTRPPL-2 antibody.

additional IFA images of the ΔTgTRPPL-2 and ΔTgTRPPL-2-trppl2 (*Figure 2—figure supplement 1B, D*). The western blots in *Figure 2—figure supplement 1B, C* showed a band at ~115 kDa in lysates of *TatiΔku80*, *TgTRPPL-2-smHA and* RH (wild type strain), which was absent in the ΔTgTRRPL-2 mutant. This band most likely represented the endogenously cleaved TgTRPPL-2. In the TgTRPPL2-smHA mutant lysate there was an additional band of ~150 kDa from the smHA tag, which added approximately 39 kDa to the protein, because of the smHA tag. The sum of both bands (120 + 150 kDa) represents the size predicted for TgTRPPL-2 in ToxoDB plus the smHA tag.

We next evaluated if the expression of *TgTRPPL-2* would impact *T. gondii* growth by plaque assays, in which the parasite engages in repetitive cycles of invasion, replication, and egress causing host cell lysis and formation of plaques is observed as white spots by staining with crystal violet. The ΔTgTRPPL-2 mutant formed smaller plaques compared to its parental counterpart, indicating a growth defect (*Figure 2D*). This growth defect was partially restored in the complemented cell line (*Figure 2D*). We reasoned that the overexpression of *TgTRPPL-2* in the ΔTgTRPPL-2-trppl2 mutant likely affected parasite fitness masking the rescue effect.

To determine which step of the lytic cycle was affected, we performed invasion and egress assays. For invasion, we used the red green assay (*Kafsack et al., 2004*) under two extracellular $Ca^{2+}$ (1.8 and 0.5 mM) conditions. Quantification of invasion in the presence of 1.8 mM $Ca^{2+}$ showed a lower invasion rate for the ΔTgTRPPL-2 (*Figure 2E*). Reducing the extracellular concentration of $Ca^{2+}$ to 0.5 mM resulted in a reduced rate of invasion in the parental cell line, which was similar to the invasion rate of the ΔTgTRPPL-2 mutants. This result suggested that TgTRPPL-2 is likely functional at higher concentrations of extracellular $Ca^{2+}$.

Egress of intracellular tachyzoites can be triggered by permeabilizing infected host cells with saponin in the presence of a buffer containing 1.8 mM $Ca^{2+}$. Under these conditions, egress of the ΔTgTRPPL-2 mutant was slower than egress of the parental strain (*Figure 2F*). Additionally, when egress was stimulated by the phosphodiesterase inhibitor Zaprinast, which increases cytosolic $Ca^{2+}$ by permitting increase of the cyclic nucleotide cGMP (*Brown et al., 2016*; *Sidik et al., 2016*), it also resulted in the ΔTgTRPPL-2 mutant taking longer to egress (*Figure 2G*). This delayed egress was observed with and without extracellular $Ca^{2+}$. For both assays tested, the ΔTgTRPPL-2 mutant took twice the time to egress compared to the parental line.

In summary, disruption of the TgTRPPL-2 locus negatively impacted two important steps of the *T. gondii* lytic cycle, invasion and egress, which impaired parasite growth.

## TgTRPPL-2 is important for cytosolic $Ca^{2+}$ influx

We previously showed that *T. gondii* extracellular tachyzoites allow influx of $Ca^{2+}$ when exposed to 1.8 mM extracellular $Ca^{2+}$ (*Pace et al., 2014*). To determine the role of TgTRPPL-2 in this pathway, we loaded extracellular ΔTgTRPPL-2 parasites with Fura-2-AM to study intracellular $Ca^{2+}$ changes after exposing them to 1.8 mM extracellular $Ca^{2+}$ (*Figure 3A*). The resting cytosolic $Ca^{2+}$ concentration of the ΔTgTRPPL-2 mutant was around 75 nM, which is similar to the resting concentration of parental cells (~70–100 nM). Adding 1.8 mM $Ca^{2+}$ to the extracellular buffer caused an increase in cytosolic $Ca^{2+}$ in both the parental strain and the ΔTgTRPPL-2 mutant (*Figure 3B*). However, $Ca^{2+}$ influx of the ΔTgTRPPL-2 mutant was decreased by approximately 50% (*Figure 3C*). The ΔTgTRPPL-2-trppl2 mutant, however, regained the $Ca^{2+}$ influx activity and showed higher $Ca^{2+}$ influx than parental cells, consistent with the higher expression of *TgTRPPL-2* shown by qPCR (*Figure 2B*). The

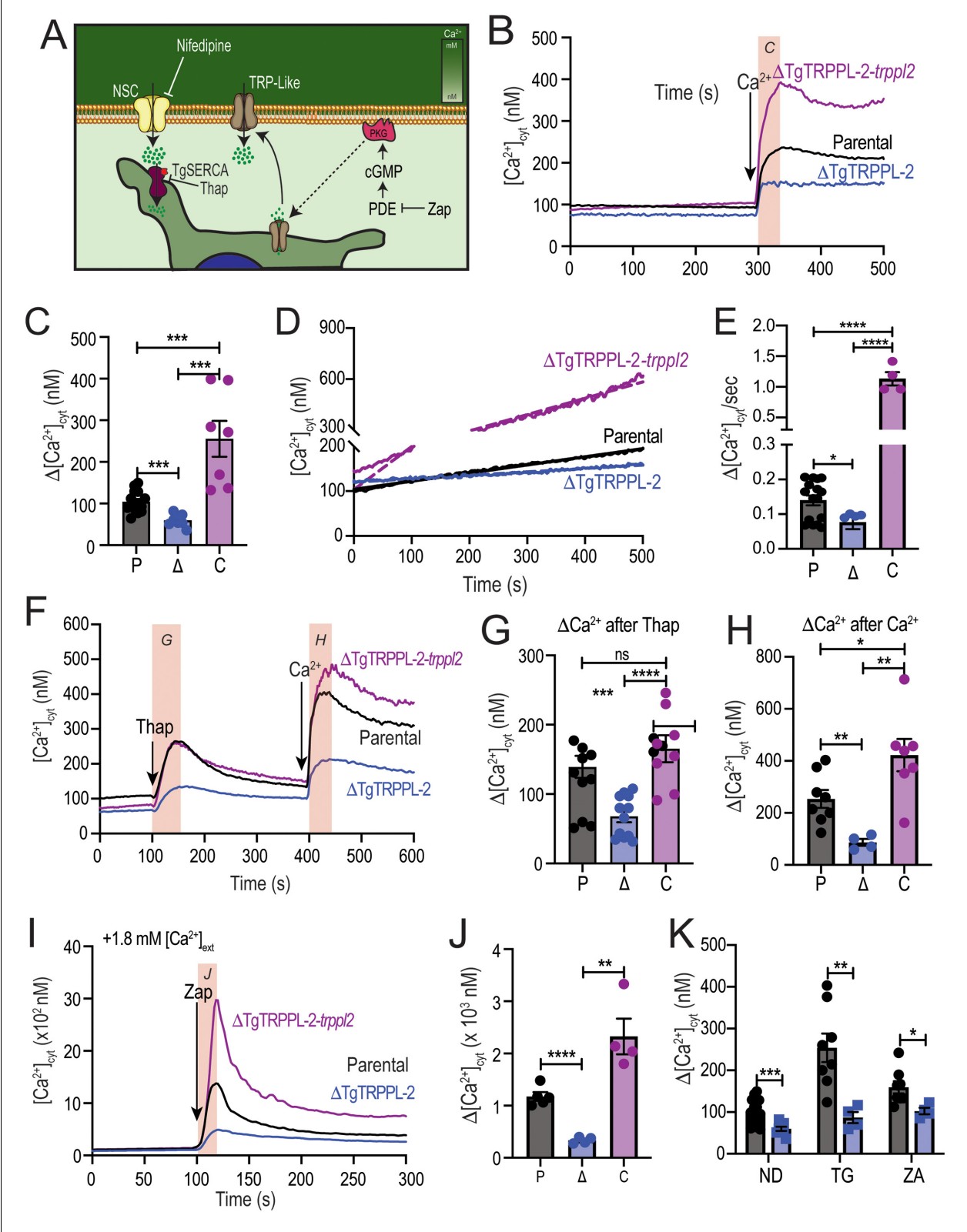

**Figure 3.** The role of TgTRPPL-2 in PM Ca²⁺ influx. (**A**) Scheme showing the mechanism of Ca²⁺ influx and how cytosolic Ca²⁺ may activate the PM channel (Ca²⁺-activated calcium entry). NSC: nifedipine-sensitive channel; PKG: protein kinase G; PDE: phosphodiesterase; Thap: thapsigargin; Zap: Zaprinast.(**B**) Cytosolic Ca²⁺ measurements of Fura-2 loaded tachyzoites of the parental (RH), Δ*TgTRPPL-2* and Δ*TgTRPPL-2-trppl2* lines. The buffer contains 100 μM ethylene glycol-bis(β-aminoethyl ether)-*N*,*N*,*N′*,*N′*-tetraacetic acid (EGTA) to chelate contaminating Ca²⁺ and at 300 s, 1.8 mM Ca²⁺

*Figure 3 continued on next page*

Figure 3 continued

were added to the suspension. The *pink box* indicates the area used for the quantification presented in (C). (C) Quantification and statistical analysis of the change in cytosolic $Ca^{2+}$ during the first 20 s after addition of extracellular $Ca^{2+}$. ***p<0.0002. (D) Constitutive $Ca^{2+}$ influx into the cytosol of parasites suspended in a buffer with 1.8 mM $Ca^{2+}$. (E) Quantification and statistical analysis of the slopes from (D). ****p<0.0001. (F) Cytosolic $Ca^{2+}$ increase after adding Thap (1 µM) followed by $Ca^{2+}$ influx after the addition of 1.8 mM extracellular $Ca^{2+}$ at 400 s. The *pink boxes* indicate the area used for the quantification presented in (G) and (H). (G) Quantification and statistical analysis of the change in cytosolic $Ca^{2+}$($\Delta[Ca^{2+}]_{cyt}$) at 50 s after the addition of Thap. (H) Quantification of the $\Delta[Ca^{2+}]_{cyt}$ 20 s after the addition of 1.8 mM of $Ca^{2+}$. ***p<0.0008, ****p<0.0001. (I) Cytosolic $Ca^{2+}$ increase stimulated by Zaprinast (100 µM) in the presence of 1.8 mM extracellular $Ca^{2+}$. (J) Quantification and statistical analysis of the $\Delta[Ca^{2+}]_{cyt}$ during the first 15 s after adding Zaprinast (100 µM) (*pink box*, in I). **p<0.001, ****p<0.0001. (K) Quantification and statistical analysis of the $\Delta[Ca^{2+}]_{cyt}$ during the 20 s after adding $Ca^{2+}$ without additions (ND) or after adding Thap or Zap. *p<0.02, **p<0.005, ***p<0.0008. Statistical analysis for all experiments was done from a minimum of three independent trials using Student's *t*-test.

The online version of this article includes the following source data and figure supplement(s) for figure 3:

**Source data 1.** Quantification and statistics of calcium measurements.
**Figure supplement 1.** TgTRPPL-2 regulates $Ca^{2+}$in *T. gondii*.
**Figure supplement 1—source data 1.** Quantification and statistics of calcium measurements.

reduction of $Ca^{2+}$ influx was further confirmed when adding 1 mM extracellular $Ca^{2+}$ to the $\Delta TgTRPPL-2$ mutant (*Figure 3—figure supplement 1A*).

When *T. gondii* extracellular tachyzoites were suspended in a high $Ca^{2+}$ buffer, a slow constitutive cytosolic influx of $Ca^{2+}$ was observed, which we attribute to leakage through a PM channel (*Figure 3D*, *parental black tracing*). Interestingly, this leakage activity was significantly reduced in the $\Delta TgTRPPL-2$ mutant (*Figure 3D, E*, *blue tracing and bar*), suggesting a role for TgTRPPL-2 in constitutive cytosolic $Ca^{2+}$ influx at the PM. Additional evidence is provided by the enhanced $Ca^{2+}$ leakage observed with the $\Delta TgTRPPL-2-trppl2$ (*Figure 3D, E*, *purple trace and bar*). The high level of $Ca^{2+}$ influx observed with the $\Delta TgTRPPL-2-trppl2$ mutant may affect parasite fitness and would explain the partial growth recovery observed with this mutant.

$Ca^{2+}$ channels may also be modulated by $Ca^{2+}$ itself (*Petri et al., 2010*). We previously showed that a cytosolic $[Ca^{2+}]$ increase may activate $Ca^{2+}$ influx at the PM ($Ca^{2+}$-activated-$Ca^{2+}$ entry) (*Pace et al., 2014*). We next investigated if TgTRPPL-2 expressed at the PM played a part in the $Ca^{2+}$-activated-$Ca^{2+}$ entry (CACE) activity. We added Thap to tachyzoites in suspension (*Figure 3F*), which resulted in a cytosolic $Ca^{2+}$ increase due to inhibition of the SERCA-$Ca^{2+}$-ATPase (SERCA) and in uncompensated $Ca^{2+}$ efflux into the cytosol. This elevated cytosolic $Ca^{2+}$ stimulated further $Ca^{2+}$ influx at the PM, which was observed as an increased rate of extracellular $Ca^{2+}$ influx (*Figure 3F*, *black trace*). The rate of $Ca^{2+}$ increase ($\Delta[Ca^{2+}]_{cyt}$) was measured as the change in the $[Ca^{2+}]_{cyt}$ following the addition of Thap (*Figure 3G*) and following the addition of $Ca^{2+}$ (*Figure 3H*). Note that the $\Delta[Ca^{2+}]_{cyt}$ shown in *Figure 3H*, *black column*, was almost 2.5 times higher than the $\Delta[Ca^{2+}]_{cyt}$ observed without previous addition of Thap (*Figure 3C*, *black column*). This CACE activity was absent in the $\Delta TgTRPPL-2$ mutant (*Figure 3F*, *blue trace*) but was restored in the $\Delta TgTRPPL-2-trppl2$-complemented strain (*Figure 3F*, *purple trace*). Quantifications of the rate of $Ca^{2+}$ increase after adding $Ca^{2+}$ and statistical analyses are shown in *Figure 3H*. Note that the $\Delta TgTRPPL-2$ mutant showed a reduced response to the addition of Thap and also to the addition of $Ca^{2+}$. Comparing the response to the addition of extracellular $Ca^{2+}$ shown in *Figure 3H*, *blue column*, the $\Delta[Ca^{2+}]_{cyt}$ is similar to the one measured directly without previous addition of Thap (compare with the *blue column* in *Figure 3C*). This result points to a complete absence of the modulatory effect of cytosolic $Ca^{2+}$ on the PM $Ca^{2+}$ influx in the $\Delta TgTRPPL-2$ mutant, which was restored in the $\Delta TgTRPPL-2-trppl2$ mutant.

We next tested the effects of Zaprinast. We previously showed that cytosolic $Ca^{2+}$ increase was almost 2.5 times higher in the presence of extracellular $Ca^{2+}$ compared with the absence of extracellular $Ca^{2+}$ (*Sidik et al., 2016*). We attributed this increase to stimulation of the PM $Ca^{2+}$ channel by cytosolic $Ca^{2+}$ (CACE). When testing CACE with the $\Delta TgTRPPL-2$ mutants, we observed that the increased response was absent (*Figure 3I–K*). Even the release of $Ca^{2+}$ from intracellular stores by Zaprinast in the presence of low extracellular $Ca^{2+}$ (~50 nM) was significantly decreased in the $\Delta TgTRPPL-2$ mutant (*Figure 3—figure supplement 1B, C*). The modulatory action of elevated cytosolic $Ca^{2+}$ in $Ca^2$ entry is shown in *Figure 3K*, which compares the $\Delta[Ca^{2+}]_{cyt}$ rate following the addition of thapsigargin (TG) or Zaprinast (ZA), which is absent in the $\Delta TgTRPPL-2$ mutant (*Figure 3I–K*).

Taken together, these results support a role for TgTRPPL-2 in Ca$^{2+}$ influx at the PM. In addition, TgTRPPL-2 is modulated by cytosolic Ca$^{2+}$ and is responsible for constitutive PM Ca$^{2+}$ influx.

## TgTRPPL-2 is a cation-conducting channel

With the aim of establishing whether TgTRPPL-2 functions as a channel and whether it is able to conduct Ca$^{2+}$, we cloned the cDNA of the *TgTRPPL-2* gene into a mammalian expression vector (pCDNA3.1) for expression in human embryonic kidney 293 cells (HEK-3KO) (*Alzayady et al., 2016*). These HEK cell line is genetically modified and the three isoforms of the inositol 1,4,5-trisphosphate receptor (IP$_3$R) are deleted to reduce background Ca$^{2+}$ currents (*Alzayady et al., 2016*). TgTRPPL-2 was mostly expressed at the ER of HEK cells as assessed by co-localization with a red fluorescent protein (mCherry) targeted to the ER and compared with the human homolog polycystin 2 (PC2) (*Figure 4A*). Because of this, we isolated nuclei (*Figure 4B*) for patch clamp experiments of the nuclear ER and further characterization of the permeability properties of TgTRPPL-2 using a modified outside-out single-channel patch clamp configuration called cytosolic-side out (*Mak et al., 2013b*; *Figure 4B, C*).

In the presence of 1.8 mM Ca$^{2+}$ inside the patch pipette and 100 nM Ca$^{2+}$ in the bath solution (see scheme of *Figure 4B*), the membranes isolated from control cells, held at –80 mV, showed very little activity and the current remained at less than 1.5 pA (*Figure 4D*, *control trace*). Some channel activity was observed after artificially depolarizing membranes (–80 to +20 mV) presumably due to opening of potassium channels. In comparison, when analyzing membranes isolated from cells expressing TgTRPPL-2 a significant increase in the open probability and current size was observed (*Figure 4D*, *TgTRPPL-2 blue trace*). The current-voltage relationship was linear and significantly different from the one from control cells (*Figure 4E*, *blue vs. black line*).

We compared the activity of TgTRPPL-2 with the human TRPP channel PC2 in parallel experiments since PC2 has been well characterized in the literature. Activity of PC2-expressing cells displayed a voltage-dependent behavior as the current-voltage relationship was not linear, with a conductance of ~73 pS (*Figure 4D, E*, *red trace*). Previous work has demonstrated that PC2 can be voltage dependent (*Kleene and Kleene, 2017*). Depending on the experimental conditions, the conductance for PC2 varied between different reports (*Kleene and Kleene, 2017*). However, comparing our experimental approach to previous work with similar experimental solutions, our calculated slope conductance for PC2 in a high Ca$^{2+}$ solution was comparable (~73 pS vs. ~97 pS). In contrast to human PC2, TgTRPPL-2 does not appear to be voltage dependent, and the conductance (55 pS) of the channel was similar to the human homologue.

Polycystin TRP channels have been shown to be activated by Ca$^{2+}$ due to the presence of an EF-hand motif at the C-terminus (*Celić et al., 2008*; *Petri et al., 2010*). To determine whether Ca$^{2+}$ can modulate the activity of TgTRPPL-2, we varied the [Ca$^{2+}$] inside the pipette (i.e., the luminal side). When the Ca$^{2+}$ concentration was increased from 1.8 mM to 10 mM Ca$^{2+}$, there was a significant inhibition of TgTRPPL-2 channel activity. With high pipette Ca$^{2+}$ concentration, the channel displayed voltage-dependent inhibition over the −80 to −20 mV range and conductance was significantly decreased (*Figure 4F*). The conductance decreased to ~21 pS when Ca$^{2+}$ was increased to 10 mM.

Although no evidence for a conserved EF-hand motif was found in TgTRPPL-2, we checked for the potential modulation by cytosolic Ca$^{2+}$. Increasing the concentration of Ca$^{2+}$ in the bath solution from 100 nM to 10 μM (which would simulate changes in cytosolic Ca$^{2+}$) enhanced channel activity from membranes expressing TgTRPPL-2 (*Figure 4G, H*, *blue vs. gold line*). Interestingly, increasing the [Ca$^{2+}$] only increased the open probability at −80 mV (*Figure 4I*). However, increase of the [Ca$^{2+}$] of the bath solution increased the conductance of the channel almost 2.5×, suggesting modulation of the channel by Ca$^{2+}$ itself. These data indicate that TgTRPPL-2 was able to conduct Ca$^{2+}$ currents and was modulated by cytosolic Ca$^{2+}$.

To distinguish whether TgTRPPL-2 is able to conduct cation currents and to determine if the activity measured could be the result of permeation of potassium, we replaced potassium with the non-permeable ion cesium (*Almog and Korngreen, 2009*; *Wu et al., 1999*). In the presence of 1.8 mM Ca$^{2+}$ inside the pipette, in a cesium chloride solution, membranes from TgTRPPL-2 and PC2-expressing cells had a significantly higher activity than control cells (*Figure 5A*). The current-voltage relationship was linear through different applied voltages and significantly different from that of control cells in potassium or cesium chloride solution (*Figure 5B*). Although channel conductance was

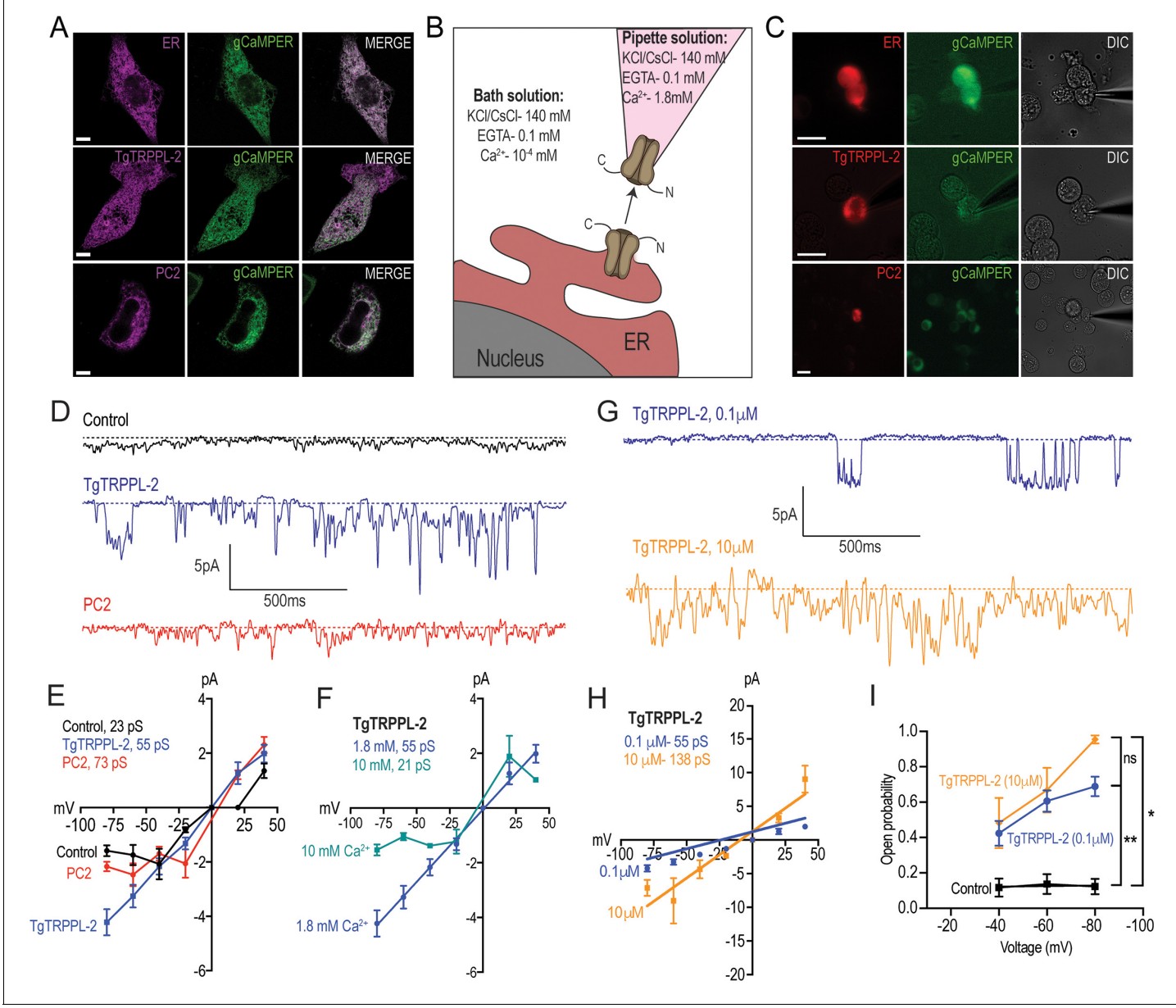

**Figure 4.** Characterization of TgTRPPL-2 expressed in HEK-3KO cells. (**A**) Images of HEK-3KO cells expressing an endoplasmic reticulum (ER)-marker, polycystin 2 (PC2), or TgTRPPL-2 with the genetic calcium indicator gCaMPer. (**B**) Schematic representation of nuclear-patch clamp in the outside (cytosolic-side) out configuration. Ionic composition and concentration for bath and pipette solutions are shown. (**C**) Patched nuclear-extract expressing ER-marker, PC2, and TgTRPPL-2 with the genetic calcium indicator gCaMPer. (**D**) Representative tracing from control, *TgTRPPL-2* or *PC2*-expressing cells showing the currents recorded in the presence of 1.8 mM luminal $Ca^{2+}$ in a symmetrical potassium chloride solution. Tracings represent approximately 2 s. (**E**) Current-voltage relationship comparing single-channel current amplitude of control, PC2, and TgTRPPL-2-expressing cells from (**D**). Inset: calculated slope conductance for control, TgTRPPL-2, and PC2. (**F**) Current-voltage relationship comparing single-channel current amplitude of TgTRPPL-2-expressing cells at 1.8 and 10 mM [$Ca^{2+}$] inside the pipette. Inset: calculated slope conductance for the conditions analyzed. (**G**) Representative traces of currents recorded from TgTRPPL-2-expressing cells using different concentration of [$Ca^{2+}$] in the bath solution (Solution A vs. Solution C) (***Supplementary file 4***). Tracings represent approximately 2 s. (**H**) Current-voltage relationship comparing single-channel current amplitude of TgTRPPL-2-expressing cells at 0.1 and 10 μM [$Ca^{2+}$] in the bath solution. Inset: calculated slope conductance for the different [$Ca^{2+}$]. (**I**) Open probability of control and TgTRPPL-2-expressing cells in the presence of different [$Ca^{2+}$] in the bath solution in comparison to the control. *$p < 0.01$, **$p < 0.001$. Scale bars in A and C represent 10 μm.

The online version of this article includes the following source data for figure 4:

**Source data 1.** Open probability measurements and statistics.

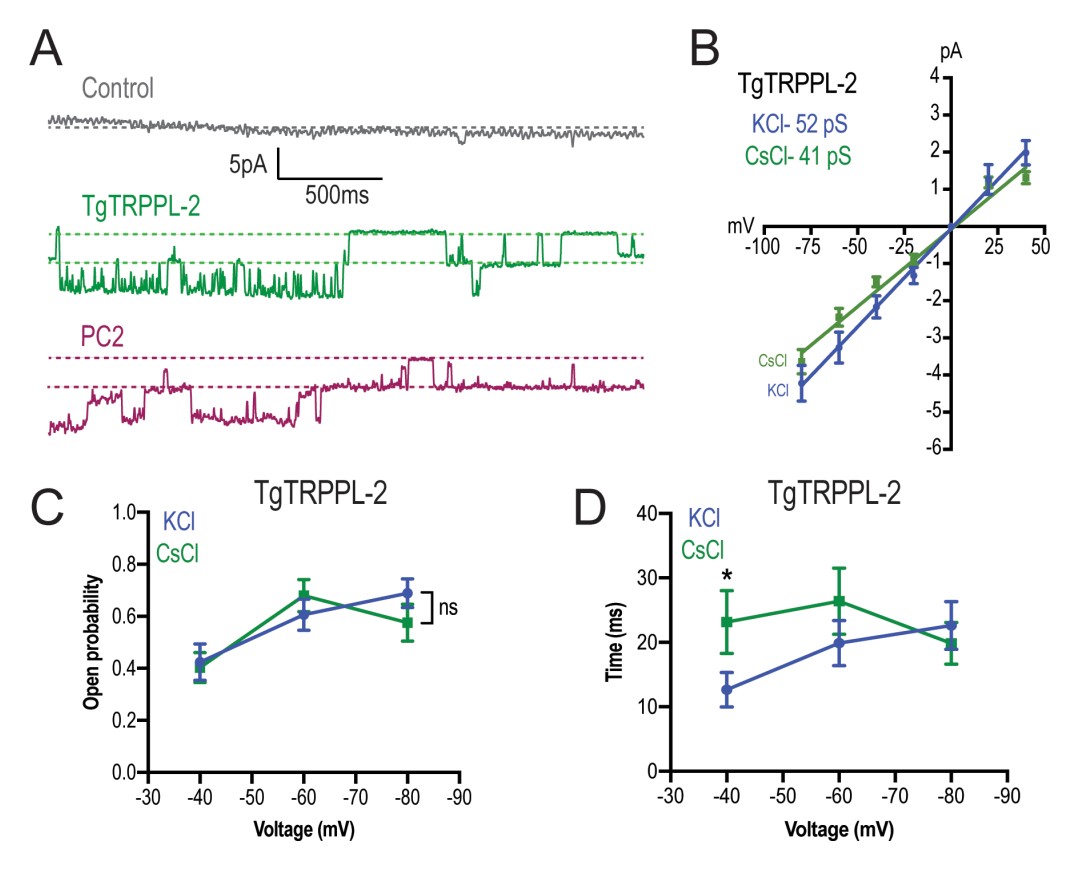

**Figure 5.** TgTRPPL-2 permeates $Ca^{2+}$. (**A**) Representative traces of currents recorded at −80 mV in the presence of 1.8 mM $Ca^{2+}$ inside the pipette (Solution D, *Supplementary file 4*) of nuclear extracts from control, TgTRPPL-2-, or polycystin 2 (PC2)-expressing cells. Traces are a representation of 2 s. (**B**) Current-voltage relationship comparing single-channel current amplitude of TgTRPPL-2 cells in 1.8 mM in KCl (*blue*) or CsCl (*green*) buffer. Inset: slope conductance of TgTRPPL-2 in the different conditions analyzed. (**C**) Calculated open probability of TgTRPPL-2-expressing cells in the presence of 1.8 mM $Ca^{2+}$ in a KCl (*blue*) or CsCl (*green*) buffer. (**D**) Average time of channel openings (dwell time) of TgTRPPL-2-expressing cells in the presence of 1.8 mM $Ca^{2+}$ in a KCl (*blue*) or CsCl (*green*). *$p<0.04$. Values are means ± SEM.

The online version of this article includes the following source data and figure supplement(s) for figure 5:

**Source data 1.** Channel amplitude measurements, open probability, and statistics.

**Figure supplement 1.** Measurement of endoplasmic reticulum (ER) calcium of HEK-3KO cells expressing TgTRPPL-2.

**Figure supplement 1—source data 1.** Calcium fluorescence measurements and statistics.

slightly higher in potassium chloride, it was not significantly different than the calculated conductance and open probability obtained in cesium chloride (*Figure 5B, C*). However, when applying voltages higher than −40 mVs, the channel was open for longer times in the presence of cesium chloride vs. potassium chloride with calcium (*Figure 5D*, *green vs. blue line*). These results indicated that TgTRPPL-2 permeates $Ca^{2+}$, however, potassium may modify the current properties.

We further demonstrated that TgTRPPL-2 was able to conduct $Ca^{2+}$ by recording $Ca^{2+}$ changes of TgTRPPL-2-HEK-3KO or mCherry-ER-HEK-3KO cells co-transfected with a genetic $Ca^{2+}$ indicator, which enabled measurement of ER luminal $Ca^{2+}$ changes and whole-ER current activity (by breaking into the ER, but not pulling the membrane away) simultaneously. Luminal $Ca^{2+}$ changes were followed through one cycle of membrane depolarization from −80 mV to 40 mV (*Figure 5—figure supplement 1A*). The fluorescence of the $Ca^{2+}$ indicator decreased in the TgTRPPL-2-expressing cells with time as voltage was applied. In both potassium as well as cesium chloride solutions with 1.8 mM $Ca^{2+}$ in the pipette, we observed that the fluorescence decrease was significantly larger when the HEK-3KO cells expressed *TgTRPPL-2* (*Figure 5—figure supplement 1B, C* vs. D, E). The slope for the fluorescence decrease appeared higher in the cesium chloride solution than in the potassium solution, although was quite variable (*Figure 5—figure supplement 1F, G*). In summary, the

observed decrease in the fluorescence of the $Ca^{2+}$ indicator supported the $Ca^{2+}$ permeation activity of TgTRPPL-2, which agrees with the single-channel conductance measurements.

## TgTRPPL-2 can be inhibited by TRP channel inhibitors and is functional at the membrane of the ER

Taking into account that TgTRPPL-2 is a cation-permeable channel and partially localizes to the periphery (*Figure 6A*), we next investigated if the residual $Ca^{2+}$ influx activity observed with the *ΔTgTRPPL-2* mutant was sensitive to anthranilic acid (ACA), a wide spectrum TRP channel inhibitor (*Harteneck et al., 2007*). ACA inhibited $Ca^{2+}$ influx by 40–50% of the parental cell line (*Figure 6B*, *black vs. red traces*). However, preincubation of the *ΔTgTRPPL-2* mutant with ACA did not further reduce $Ca^{2+}$ influx (*Figure 6B*, *dark blue vs. light blue* and 6C, *blue points*). We previously reported that $Ca^{2+}$ influx in *T. gondii* was inhibited by L-type voltage-gated $Ca^{2+}$ channel blockers like nifedipine (*Pace et al., 2014*). Preincubation with nifedipine inhibited $Ca^{2+}$ influx of the parental cell line by almost 80% (*Figure 6D*, *black vs. green bar* in P). The rate of $Ca^{2+}$ influx, defined as the $\Delta Ca^{2+}$ immediately following the addition of $Ca^{2+}$, of the *ΔTgTRPPL-2* mutant was inhibited by nifedipine by almost 100% (*Figure 6D*, *black vs. green bar* in $\Delta$). Rates of $\Delta Ca^{2+}$ after adding extracellular $Ca^{2+}$ and its inhibition by ACA and nifedipine for parental (P), *ΔTgTRPPL-2* ($\Delta$), and *ΔTgTRPPL-2-trppl2* (C) mutants are shown in *Figure 6D*. The $\Delta Ca^{2+}$ was highest for the complemented mutant as TgTRPPL-2 was overexpressed but the % of inhibition by ACA was still around 50% (*Figure 6D*, *black vs. purple bar* in C). These results point to TgTRPPL-2 functioning as a $Ca^{2+}$ conducting channel that is important for $Ca^{2+}$ influx from the extracellular milieu and is sensitive to TRP channel inhibitors.

The localization of TgTRPPL-2 at the ER of tachyzoites (*Figure 1D*) indicated its potential function in cytosolic $Ca^{2+}$ efflux observed after inhibiting TgSERCA with Thap (*Moreno and Zhong, 1996*). ER efflux activity observed after addition of Thap was significantly decreased by preincubation of Fura-2-loaded wild-type tachyzoites with ACA (*Figure 6E*, *black vs. red traces*). Interestingly, ACA inhibited the rate of ER $Ca^{2+}$ efflux, which was comparable to the reduced efflux rate triggered by Thap of the *ΔTgTRPPL-2* mutant (*Figure 6F*, *compare red and blue bar*).

We next investigated whether ACA inhibited the previously described CACE mechanism (*Pace et al., 2014*). We first tested the parental strain by preincubating parasites with ACA, adding Thap at 100 s followed by the addition of extracellular $Ca^{2+}$ as indicated (*Figure 6E–G*). Example traces demonstrating $Ca^{2+}$ entry at the PM by elevated cytosolic $Ca^{2+}$ are shown at two different times after the addition of Thap, 50 s (*Figure 6G*) and 300 s (*Figure 6E*). The analysis of the rates is shown in *Figure 6H*, and the comparison of the $\Delta[Ca^{2+}]$/s of $Ca^{2+}$ entry without previous addition of Thap (*Figure 6H*, *black bar*) with the rate at 50 s and at 300 s after Thap (*Figure 6H*, *green and maroon*). This stimulation by cytosolic $Ca^{2+}$ was abolished by ACA (*Figure 6E*, *black vs. red trace* and 6H, *gold and red bars*). $Ca^{2+}$ entry following cytosolic $Ca^{2+}$ increase by Thap addition was also reduced to the basal rate in the *ΔTgTRPPL-2* mutant (*Figure 6H*, *blue bar*). The modulation of the $Ca^{2+}$ entry mechanism by cytosolic $Ca^{2+}$ was lost in the *ΔTgTRPPL-2* mutant (*Figure 6H*, *black vs. blue bars*). These data suggest that ACA inhibited both efflux of $Ca^{2+}$ from the ER as well as $Ca^{2+}$-induced $Ca^{2+}$ entry. This led us to propose that TgTRPPL-2, in addition to mediating $Ca^{2+}$ entry at the plasma membrane, may also mediate $Ca^{2+}$ efflux/leakage from the ER, a pathway sensitive to the TRP channel inhibitor ACA.

To further validate the specificity of ACA for the inhibition of TgTRPPL-2, we tested the ability of this inhibitor and a second broad-spectrum TRP channel inhibitor, benzamil, to inhibit single-channel current activity. The channel activity of TgTRPPL-2 was significantly decreased by both ACA and benzamil (*Figure 7A*). ACA diminished the amplitude of the current, reduced the open probability, and the time that the channel remained open (dwell time) (*Figure 7B*). Of the current that remained in the presence of ACA, the conductance was reduced to almost half (*Figure 7B*). The inhibition of the open probability fits with the inhibition of $Ca^{2+}$ entry in *T. gondii*. In comparison to ACA, benzamil only reduced the open probability of TgTRPPL-2 but not the length of time the channel was open (*Figure 7C, D*). The conductance of TgTRPPL-2 was reduced to one third of the control in the presence of benzamil.

Based on the pharmacological results, we predicted that the TRP inhibitors should also inhibit in vitro growth assays (*Figure 7E, F*, *top panel and parental bars*). Both ACA and benzamil significantly inhibited in vitro *T. gondii* growth. We calculated the $IC_{50}$ for ACA at 1.4 ± 0.4 µM. Consistent with ACA and benzamil targeting TRP channels, neither drug affected the already attenuated growth of

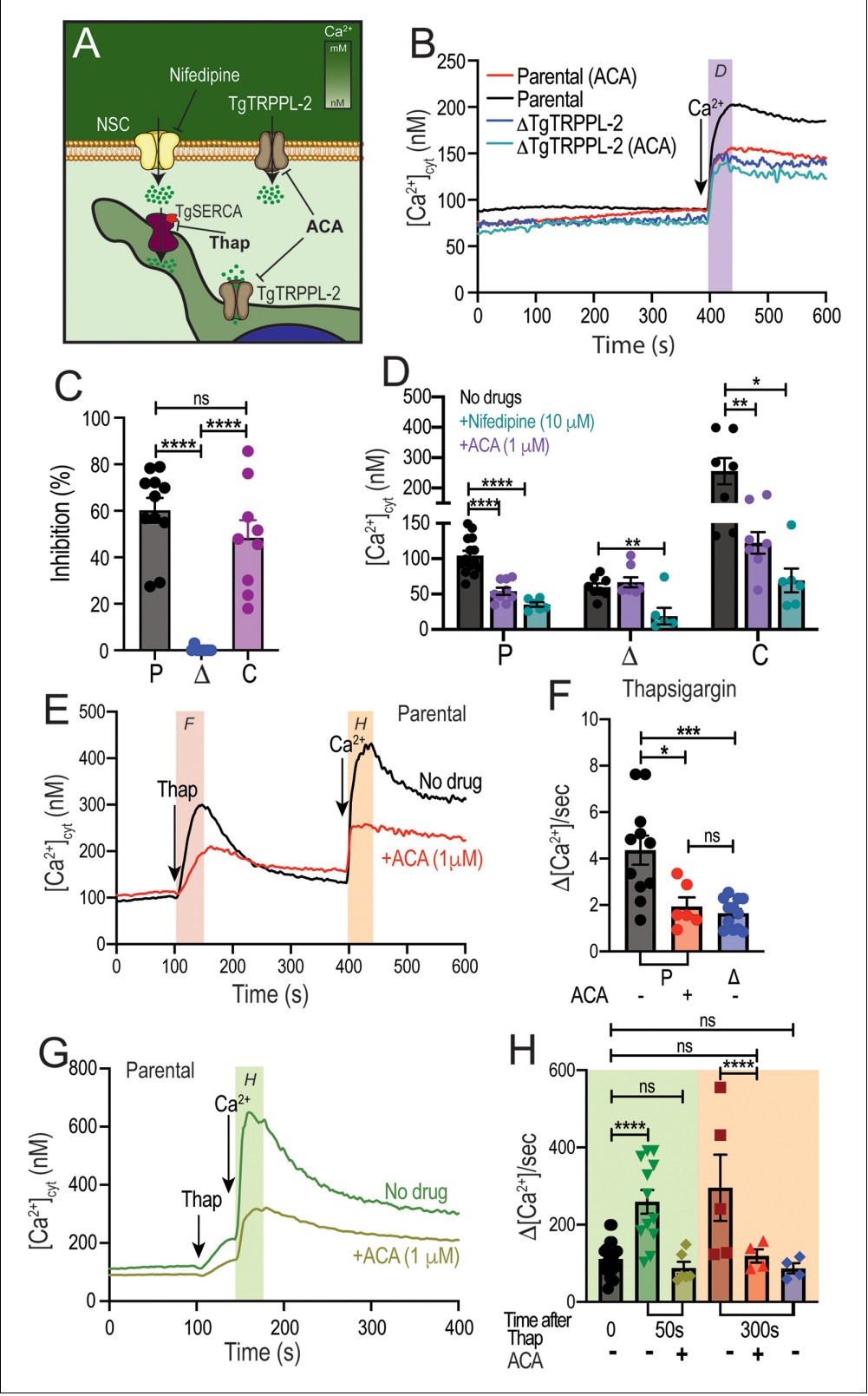

**Figure 6.** Regulation of TgTRPPL-2 by $Ca^{2+}$ and inhibition by transient receptor potential inhibitors. (**A**) Scheme showing TgTRPPL-2 at the PM and endoplasmic reticulum. (**B**) Cytosolic $Ca^{2+}$ measurements of Fura-2-loaded tachyzoites preincubated with 1 µM anthranilic acid (ACA). 1.8 mM $Ca^{2+}$ was added where indicated. The *purple box* indicates the area used for the quantification presented in (**D**). (**C**) Percentage inhibition of $Ca^{2+}$ influx in the

*Figure 6 continued on next page*

*Figure 6 continued*

presence of 1 μM of ACA: P: parental, Δ: Δ*TgTRPPL-2* and C: Δ*TgTRPPL-2-trppl2*. (**D**) Change in cytosolic $Ca^{2+}$ during the first 20 s after addition of $Ca^{2+}$ in the presence of 10 μM of nifedipine or 1 μM ACA. P: parental, Δ: Δ*TgTRPPL-2* and C: Δ*TgTRPPL-2-trppl2*. *p<0.01, **p<0.003, ****p<0.0001. (**E**) Cytosolic $Ca^{2+}$ increase after adding Thap (1 μM) to a suspension of wild-type tachyzoites (RH). The red line shows a similar experiment, but the cells were preincubated with 1 μM ACA for 3 min. The *pink and orange boxes* show the areas used for the quantifications presented in (**F**) and (**H**). (**F**) Quantification and statistical analysis of the slope 50 s after the addition of Thap in the presence or absence of ACA in parental (P) and the Δ*TgTRPPL-2* mutant (Δ). *p<0.01, ***p<0.0003. (**G**) Stimulation of $Ca^{2+}$ influx 50 s after addition of Thap in the presence or absence of 1 μM ACA. The green box shows the area used for the quantifications presented in (**H**). (**H**) Quantification of change of cytosolic $Ca^{2+}$ 20 s after the addition of 1.8 mM $Ca^{2+}$ following the addition of Thap under different conditions. ****p<0.00001. The statistical analysis for all experiments was done from at least three independent trials using Student's *t*-test. Values are means ± SEM.

The online version of this article includes the following source data for figure 6:

**Source data 1.** Quantification and statistics of calcium measurements.

---

the Δ*TgTRPPL*-2 mutant (*Figure 7F*). Cilnidipine, a voltage-gated $Ca^{2+}$ channel blocker, completely inhibited growth of both parental and the Δ*TgTRPPL*-2 mutant, suggesting that the activity of the L-type $Ca^{2+}$ channel contributes to the residual growth of the Δ*TgTRPPL*-2 mutant (*Figure 7F*).

In conclusion, TgTRPPL-2 is a cation-permeable channel that can be inhibited by broad-spectrum TRP channel inhibitors. Inhibition of channel activity inhibited parasite growth.

## Discussion

In this study, we report the presence and functional role of a *T. gondii* channel, TgTRPPL-2, that localizes to the endoplasmic reticulum and plasma membrane. The corresponding gene *TgGT1_310560* was annotated as hypothetical but was predicted as a TRP channel based on a bioinformatic analysis of the *T. gondii* genome comparing it with TRP channel genes of mammalian origin (*Prole and Taylor, 2011*). Here, we established that TgTRPPL-2 is important for both $Ca^{2+}$ entry at the PM and $Ca^{2+}$ efflux from the ER of *T. gondii* tachyzoites. TgTRPPL-2, expressed in HEK-3KO cells, conducted currents in high calcium solutions and was not voltage dependent. Interestingly, $Ca^{2+}$ itself modulated the conductance of TgTRPPL-2. Broad-spectrum TRP channel inhibitors like ACA and benzamil inhibited the activity of TgTRPPL-2, $Ca^{2+}$ influx in the parasite as well as parasite growth.

Silencing of TgTRPPL-2 in the Δ*TgTRPPL-2* mutant impacted both invasion and egress of *T. gondii*, resulting in a general growth defect. Extracellular tachyzoites, which are surrounded by high $Ca^{2+}$, are able to use $Ca^{2+}$ influx to stimulate invasion of a new host cell and carry on their lytic cycle. The Δ*TgTRPPL-2* mutant showed a reduction in their host invasion ability, suggesting the defect may be due to a reduction in $Ca^{2+}$ influx because of the absence of TgTRPPL-2. Interestingly, the reduction in $Ca^{2+}$ influx (~50%) in the Δ*TgTRPPL-2* mutant was comparable to the reduction of invasion, suggesting that TgTRPPL-2 may be involved in the $Ca^{2+}$ influx pathway that stimulates invasion. Delay in the ability of the Δ*TgTRPPL-2* mutant to egress could be caused by a defective efflux of $Ca^{2+}$ from the ER, which was significantly lower in the mutant. This was evidence for the function of TgTRPPL-2 as a $Ca^{2+}$ channel at the ER membrane.

The impact of silencing TgTRPPL-2 on *T. gondii* growth was not total, and parasites were still able to perform lytic cycle activities at a reduced rate. The main defects of the Δ*TgTRPPL-2* mutant – invasion, egress, $Ca^{2+}$ influx, and ER $Ca^{2+}$ efflux – were not complete likely because more than one mechanism or channel is functional at both locations (PM and ER). We hypothesize the presence of another channel at the PM, likely the one responsible for the $Ca^{2+}$ influx activity that is inhibited by nifedipine (*Pace et al., 2014*). It is also possible that a release channel responsive to $IP_3$ may be involved in release of $Ca^{2+}$ from the ER (*Lovett et al., 2002*) with TgTRPPL-2 having a role in constitutive efflux and protecting the ER against $Ca^{2+}$ overload.

Numerous observations in *T. gondii* have demonstrated that intracellular $Ca^{2+}$ oscillations in the parasite precede the activation of distinct steps of the lytic cycle (*Hortua Triana et al., 2018b*; *Lourido and Moreno, 2015*). Influx of both extracellular and intracellular $Ca^{2+}$ pools into the

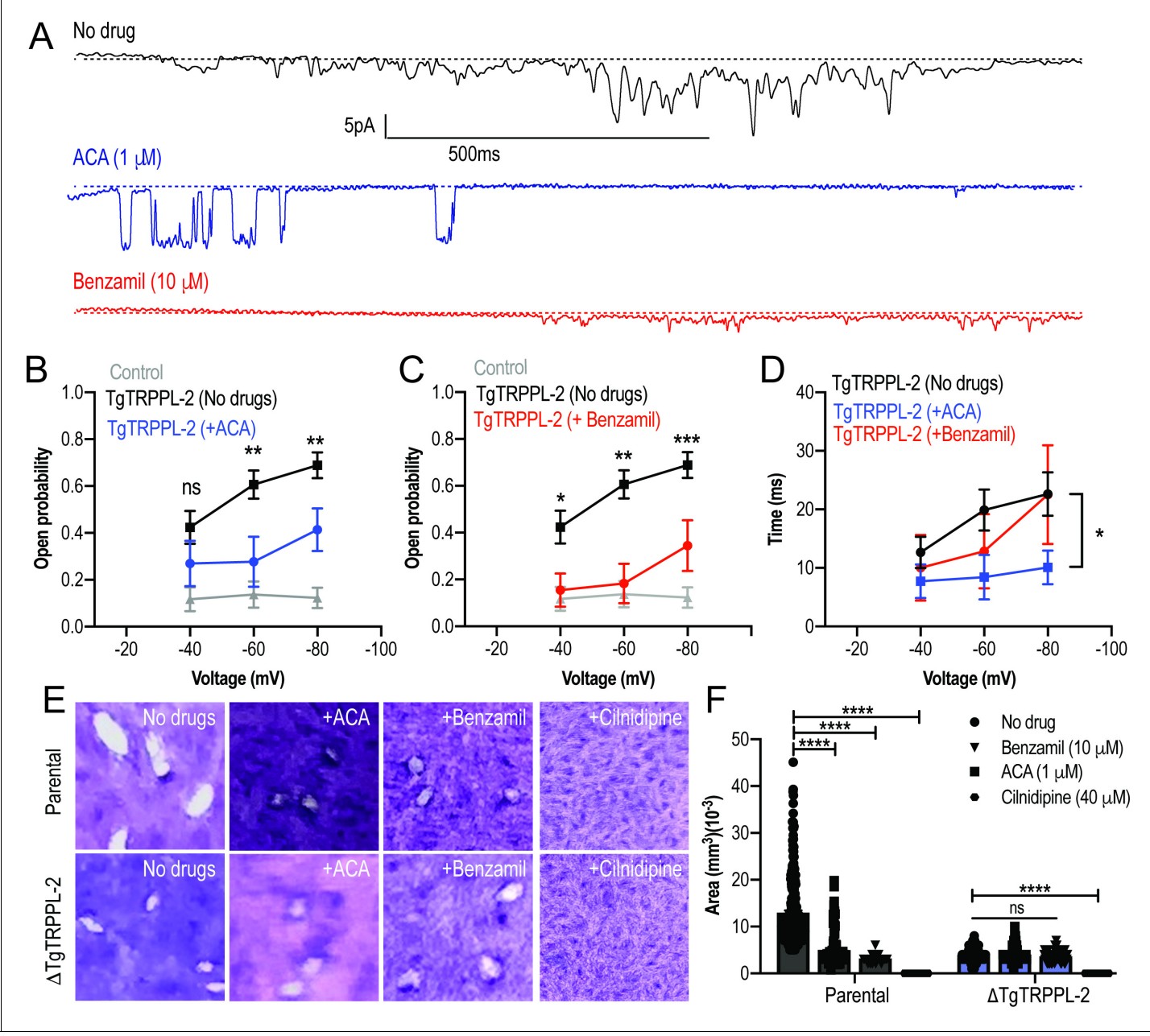

**Figure 7.** Transient receptor potential (TRP) inhibitors decreased the activity of TgTRPPL-2. (**A**) Example of currents recorded of TgTRPPL-2-expressing cells at –80 mV without inhibitors (*black trace*) compared with the currents in the presence of 1 µM of anthranilic acid (ACA) (*blue trace*) or 10 µM of benzamil (*red trace*). (**B**) Calculated open probability of TgTRPPL-2-expressing cells (*black*) or in the presence of ACA (*blue*). \*\*p<0.006–0.007. (**C**) Calculated open probability of TgTRPPL-2-expressing cells (*black*) or in the presence of benzamil (*red*). Asterisks indicate p-values for significance. \*p<0.02, \*\*p<0.002, \*\*\*p<0.0002. (**D**) Average time of channel opening (dwell time) of TgTRPPL-2-expressing cells in the presence of TRP inhibitors. Asterisks indicate p-values for significance, \*p<0.02. (**E**) Plaque assay of the Δ*TgTRPPL-2* mutant and the parental strain in the presence of ACA (1 µM), benzamil (10 µM), and cilnidipine (40 µM) after 7 days of growth. (**F**) Statistical analysis of plaque sizes done from three independent biological replicates using Student's *t*-test. Values are means ± SEM, \*\*\*\*p<0.0001.

The online version of this article includes the following source data for figure 7:

**Source data 1.** Inhibition measurements.

parasite cytosol contributes to the activation of downstream signaling pathways decoded into critical biological steps of the parasite lytic cycle (*Hortua Triana et al., 2018b*; *Lovett and Sibley, 2003*). $Ca^{2+}$ influx at the plasma membrane of *T. gondii* is highly regulated, stimulated by cytosolic $Ca^{2+}$,- and is operational in extracellular (*Pace et al., 2014*) and intracellular replicating tachyzoites (*Vella et al., 2021*). Our data with the *ΔTgTRPPL-2* mutant identified TgTRPPL-2 as a functional protein at the PM and the ER and in both locations would allow $Ca^{2+}$ influx into the cytosol. The dual localization of TgTRPPL-2 is in accord with other TRP channels in other cells, which shows a dynamic localization between vesicular organelles and the plasma membrane where they facilitate $Ca^{2+}$ influx (*Bezzerides et al., 2004*). In this regard, the human ortholog, PC2, localizes to both the plasma membrane and the ER (*Cai et al., 1999*).

*T. gondii* expresses a SERCA-$Ca^{2+}$-ATPase, a P-type ATPase, that couples ATP hydrolysis to the transport of ions across biological membranes (TgSERCA) and localizes to the ER (*Nagamune et al., 2007*). TgSERCA is sensitive to Thap, a sesquiterpene lactone derived from the plant *Thapsia garganica* (*Sagara and Inesi, 1991*; *Thastrup et al., 1990*), and previous studies showed that inhibition of TgSERCA by Thap resulted in cytosolic $Ca^{2+}$ efflux through an unknown channel (*Moreno and Zhong, 1996*; *Pace et al., 2014*). In mammalian cells, the passive $Ca^{2+}$ efflux from the ER is thought to prevent ER $Ca^{2+}$ overload and help maintain the steady-state concentration of luminal $Ca^{2+}$ permitting cytosolic $Ca^{2+}$ signaling (*Clapham, 2007*; *Guerrero-Hernandez et al., 2010*). Several membrane proteins have been proposed to be involved in the ER $Ca^{2+}$ efflux/leak pathway including TRP channels (*Carreras-Sureda et al., 2018*). Results from this work support a role for TgTRPPL-2 in ER $Ca^{2+}$ efflux in *T. gondii* because $Ca^{2+}$ efflux from the ER caused by the addition of Thap or Zaprinast was significantly decreased in the *ΔTgTRPPL-2* mutants. These results support a functional role for TgTRPPL-2 at the membrane of the ER as the constitutive leak channel involved in $Ca^{2+}$ efflux when the store is filled. This could also be the mechanism by which the ER supplies $Ca^{2+}$ to other organelles like the mitochondria or the plant-like vacuole (PLV), a lysosome-like compartment that stores $Ca^{2+}$ (*Miranda et al., 2010*).

Previous work from our laboratory showed that $Ca^{2+}$ influx at the plasma membrane does not operate as store-operated $Ca^{2+}$ entry (SOCE), which was shown with experiments testing surrogate ions like $Mn^{2+}$ (*Pace et al., 2014*). This result was supported by the lack of components of the SOCE pathway, STIM, and ORAI from the *T. gondii* genome (*Prole and Taylor, 2011*). However, $Ca^{2+}$ influx was modulated by cytosolic $Ca^{2+}$ (*Pace et al., 2014*), and this modulation was absent in the *ΔTgTRPPL-2* mutant supporting a role for TgTRPPL-2 as the channel responsible for $Ca^{2+}$ influx at the PM activated by cytosolic $Ca^{2+}$. TRP channels have been shown to play a role in $Ca^{2+}$-activated-$Ca^{2+}$ entry (*Ta et al., 2020*). Release of $Ca^{2+}$ from intracellular stores like the ER is also significantly diminished in the *ΔTgTRPPL-2* mutant, which could affect the stimulation of $Ca^{2+}$ influx. However, when using Zaprinast, which raised cytosolic $Ca^{2+}$ at a much higher level than Thap, the stimulation of $Ca^{2+}$ entry by cytosolic $Ca^{2+}$ was absent. This further supports that TgTRPPL-2 functions at the PM-mediating $Ca^{2+}$ entry and is modulated by cytosolic $Ca^{2+}$.

We showed that TgTRPPL-2 was able to conduct currents with conductance values comparable to the values of human TRP channels (*Kleene and Kleene, 2017*; *Liu et al., 2018b*; *Vien et al., 2020*). Previous work with PC2 and other polycystin-like proteins showed that $Ca^{2+}$ modulated the activity of these proteins (*DeCaen et al., 2016*; *Kuo et al., 2014*; *Yang et al., 2015*; *Chen et al., 1999*; *Kleene and Kleene, 2017*). Sustained cytosolic $Ca^{2+}$ increase inhibited PC2-Like1 currents (*DeCaen et al., 2016*) while other studies showed that cytosolic $Ca^{2+}$ increase from physiological (100 nM) to μM levels increased the activity of polycystin L and PC2, respectively (*Chen et al., 1999*; *Kleene and Kleene, 2017*). We observed some of these responses with TgTRPPL-2 as increasing $Ca^{2+}$ inside the pipette (affecting the luminal/plasma membrane side of the channel) showed a significant decrease in the currents. Comparably, increasing $Ca^{2+}$ concentration in the bath solution (cytosolic) from physiological levels to μM levels showed an increase of 2.5× in the conductance of TgTRPPL-2. Although cytosolic $[Ca^{2+}]$ is unlikely to reach those high μM levels, the potential presence of $Ca^{2+}$ microdomains at the plasma membrane or the ER membrane would result in higher concentrations of $Ca^{2+}$ at the exit of the channel due to slow diffusion of $Ca^{2+}$ ions (*Berridge, 2006*; *Burgoyne et al., 2015*; *Mulier et al., 2017*).

Because TRP channels are non-selective cation-permeable channels, they can also permeate monovalent ions like $Na^+$ or $K^+$. When symmetrical KCl was substituted for CsCl, in the presence of $Ca^{2+}$, we found that TgTRPPL-2 could still conduct cations, strongly indicating $Ca^{2+}$ permeance.

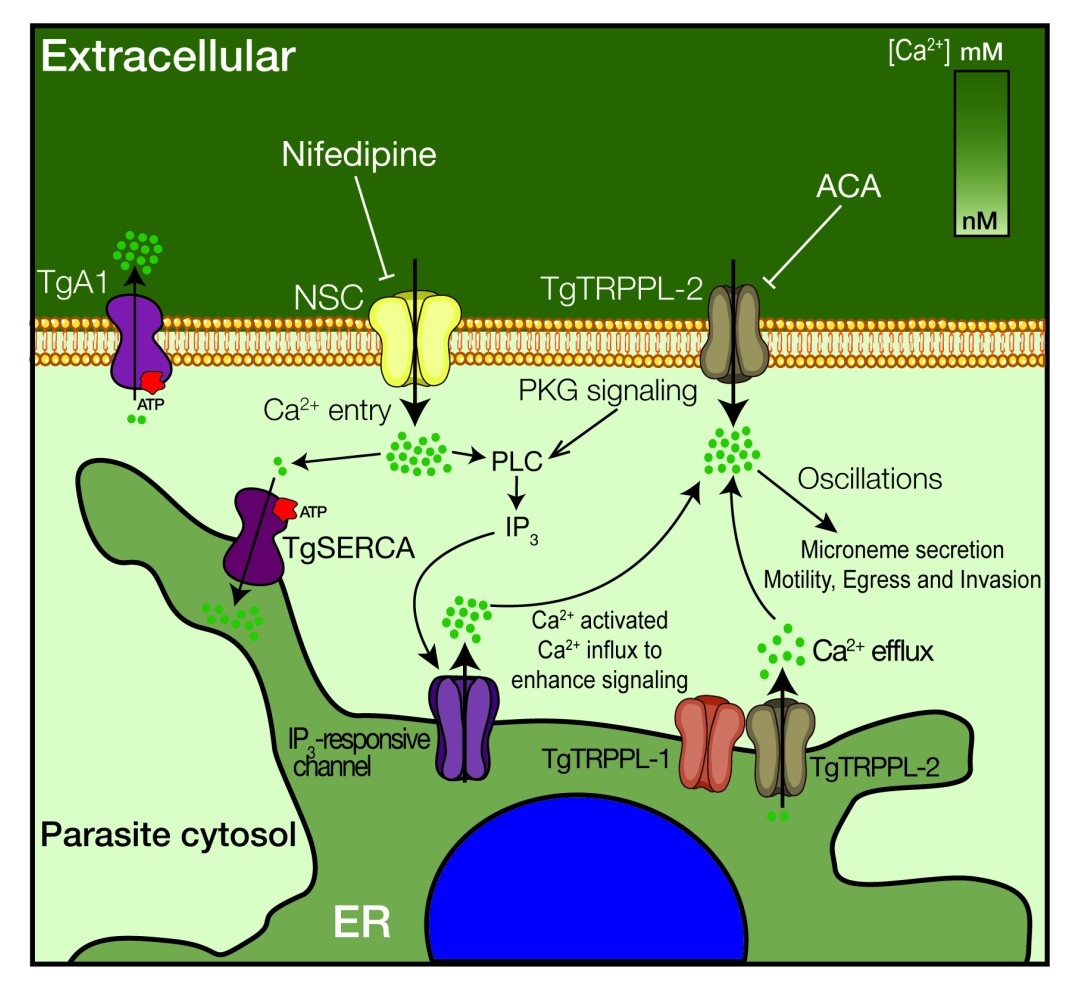

**Figure 8.** Model of the role of TgTRPPL-2 in $Ca^{2+}$ influx into the cytosol of *T. gondii*. $Ca^{2+}$ entry is mediated by two independent $Ca^{2+}$ channels at the PM, a nifedipine-sensitive channel (NSC) and TgTRPPL-2. TgTRPPL-2 is a cation-permeable channel that mediates $Ca^{2+}$ entry at the PM by a pathway that is activated by high $[Ca^{2+}]$ and can be inhibited by broad transient receptor potential inhibitors like anthranilic acid and benzamil. TgA1 is shown as the PM $Ca^{2+}$ ATPase that pumps $Ca^{2+}$ outside the cell. TgTRPPL-2 also localizes to the endoplasmic reticulum (ER) where it may function as a $Ca^{2+}$ efflux channel. Increase in cytosolic $[Ca^{2+}]$ can modulate TgTRPPL-2 by allowing the channel to open for longer time, thus allowing more $Ca^{2+}$ to enter the cell. Also shown is the IP$_3$-responsive channel in the ER. The molecular identity of this channel is not known. $Ca^{2+}$ activates the phosphatidyl inositol phospholipase C (PLC), which is also activated through protein kinase G signaling. PLC synthesizes IP$_3$, which stimulates $Ca^{2+}$ release from the ER.

However, we did not determine the ionic preference or selectivity of TgTRPPL-2, thus we can only propose that TgTRPPL-2 is a cation-permeable channel capable of conducting $Ca^{2+}$. In cilia, PKD channels have been described to have relatively high conductance (*Kleene and Kleene, 2017*; *Liu et al., 2018a*). The conductance calculated for TgTRPPL-2 is within range of what has been described for PC2 in other cells (30–157 pS). However, it is important to note that the properties described for any channel will depend on the experimental approaches used.

ACA and benzamil are broad-spectrum inhibitors that have the ability to inhibit TRP channel activity. ACA is a weak base that inhibits currents mediated by TRP channels (*Harteneck et al., 2007*). ACA does not block the pore of the channel as most inhibitors but rather reduce the open probability of the channel (*Harteneck et al., 2007*). In a similar manner, benzamil is also able to inhibit currents mediated by TRP channels by binding to a site that modulates their activity rather than blocking its pore (*Dai et al., 2007*). In our experiments testing ACA and benzamil, we observed that while the inhibitors affected $Ca^{2+}$ influx and growth of the parental cell line, neither affected the already reduced growth and $Ca^{2+}$ influx of the $\Delta TgTRPPL-2$ mutant. This result, combined with the

inhibition of TgTRPPL-2 currents impacting both open probability and time that the channel remained open, points to TgTRPPL-2 as a target of ACA and benzamil.

Recent studies on $Ca^{2+}$ signaling in *T. gondii* have expanded our understanding of the link between $Ca^{2+}$ and critical facets of parasite biology (i.e., gliding motility, microneme secretion, host cell invasion, and egress) (*Borges-Pereira et al., 2015*; *Brown et al., 2016*; *Stewart et al., 2017*). However, important molecular players have remained enigmatic, like the PM channels responsible for $Ca^{2+}$ influx and the ER channel responsible for the passive leakage into the cytosol. Characterization of TgTRPPL-2 and its function at the ER and PM fills a small gap in our knowledge of $Ca^{2+}$ signaling and homeostasis in *T. gondii* (*Figure 8*). This study is the first biophysical characterization of a channel in *T. gondii*, and TgTRPPL-2 represents the first identified molecule to mediate $Ca^{2+}$ influx into the cytosol of *T. gondii* at the plasma membrane and the ER. In addition, this study identifies TgTRPPL-2 as a potential target for combatting toxoplasmosis.

# Materials and methods

## Key resources table

| Reagent type (species) or resource | Designation | Source or reference | Identifiers | Additional information |
|---|---|---|---|---|
| Antibody | Mouse- αTgTRPPL-2 | This work | Silvia Moreno (University of Georgia) | Recognizes TgGT1_310560 in *T. gondii* tachyzoites IFA (1:100) WB (1:1000) |
| Antibody | Rabbit-αSag1 | *Mineo et al., 1993* | | IFA (1:1000) WB (1:1000) |
| Antibody | Rabbit-αSERCA | *Nagamune et al., 2007* | | IFA (1:1,000) WB (1:1000) |
| Antibody | Rabbit-αGap45 | *Gaskins et al., 2004* | | IFA (1:1000) WB (1:1000) |
| Antibody | Mouse-αTubulin | Sigma Aldrich | Cat#11867423001 | WB (1:10,000) |
| Antibody | Mouse-αHA | Roche | Cat#11867423001 | IFA (1:1000) |
| Antibody | Alexa Fluor 488 | Thermo Fisher Scientific | Cat#A20181 | IFA (1:1000) |
| Antibody | Alexa Fluor 546 | Thermo Fisher Scientific | Cat#A20183 | IFA (1:1000) |
| Cell line | ΔTgTRPPL-2 | This work | Silvia Moreno (University of Georgia) | Clonal cell line with the TgGT1_310560 disrupted Validated through genomic amplification, qPCR, IFAs, and WB |
| Cell line | TgTRPPL-2-smHA | This work | Silvia Moreno (University of Georgia) | Clonal cell line with the TgGT1_310560 gene locus tagged Validated through genomic amplification, IFAs, and WB |
| Cell line | ΔTgTRPPL-2-trppl2 | This work | Silvia Moreno (University of Georgia) | Clonal cell line with the TgGT1_310560 disrupted and expressing an extra copy of the same gene Validated through genomic amplification, qPCR IFAs, and WB |
| Cell line | RHTatiΔku80 | *Sheiner et al., 2011* | | Parental line used for tagging |
| Recombinant DNA reagent | TgTRPPL-2-smHA | This work | | Plasmid used to C-terminally tag TgGT1_310560 |
| Recombinant DNA reagent | pMOS003-lenti-CMV-gCaMPer | *Henderson et al., 2015* | RRID:Addgene #65227 | |

### *Toxoplasma* growth

All parasite strains were maintained in vitro by serial passage in Dulbecco's modified minimal essential media (DMEM) with 1% FBS, 2.5 µg/ml amphotericin B, 100 µg/ml streptomycin in the human telomerase reverse transcriptase immortalized foreskin fibroblasts (hTERT) (*Farwell et al., 2000*). hTert cells are periodically tested and treated for *Mycoplasma* contamination.

## Generation of mutants

The smHA-LIC-CAT plasmid was used for in situ C-terminal tagging of TgTRPPL-2-smHA (*Hortua Triana et al., 2018a*). Carboxy-terminus tagging was done in the parental line RHTatiΔku80 (TatiΔku80) (*Sheiner et al., 2011*), a parasite line that contains the tetracyclin-regulated transactivator system that allows conditional expression of genes (*Meissner et al., 2001*) and also in which the *ku80* gene was deleted, increasing efficiency of homologues recombination (*Fox et al., 2009*). Briefly, a homology region of 974 bp covering the 3′ region of the gene of interest excluding the STOP codon was amplified by PCR using *T. gondii* RH genomic DNA as template and cloned into the plasmid. Plasmids were validated by restriction digest and sequencing. The oligonucleotides primers used for PCR and for creating the gene-tagging plasmids and for PCR validations are listed in *Supplementary file 3* (primers T1–T3). Prior to transfection, all plasmids were linearized within the region of homology. Approximately 20 µg of plasmid DNA was used for transfection of $1 \times 10^7$ *T. gondii* RHTatiΔKu80 parasites using a Gene Pulser X Cell electroporator (BioRad). Selection was done with 20 µM chloramphenicol, and clones were isolated by limiting dilution. DNA of selected clones were isolated and screened by PCR.

To disrupt the *TgTRPPL-2* (TgGT1_310560) gene, a single-guide RNA against TgTRPPL-2 was constructed as described (*Shen et al., 2017*). The single-guide RNA was mutagenized with the desired sequence in a plasmid that contains the Cas9 using the Q5 Mutagenesis Kit following the manufacturer's instructions. The correct mutation was verified by sequencing. The pyrimethamine-resistant DHFR cassette was amplified by PCR with primers containing 50 bp homology arms of the region upstream and downstream of the start and stop codon of the *TgTRPPL-2* gene. The created sgTgTRPPL-2 CRISPR plasmid was co-transfected with the DHFR cassette (3:1, respectively) into RH tachyzoites. Selection followed with pyrimethamine for 7 days. Parasites were sub-cloned by limiting dilution, and screening for clones was done by PCR. The primers used for the creation of the ΔTgTRPPL-2 are listed in *Supplementary file 3* (primers K1–K4).

## Quantitative PCR

Total RNA from parental, Δ*TgTRPPL-2* and Δ*TgTRPPL2-trppl2* was extracted and reversed-transcribed into cDNA. The qPCR reaction was done using the iQSYBR Green master mix (BioRad), plus primers, and the reverse-transcribed cDNA (primers shown in *Supplementary file 3*, Q1–Q2). The qRT-PCR was carried out on a CFX96 PCR Real-Time detection system (C1000Touch Thermal cycler, BioRad). Relative quantification software (CFX Maestro software) was used for the analysis, and relative expression levels were calculated as the fold change using the formula $2^{ΔΔCT}$ (*Livak et al., 1995*). Normalization was done using Actin and Tubulin primers. Experiments were repeated three times with triplicate samples.

## Antibody production of TgTRPPL-2

The antigenic region for TgTRPPL-2 chosen for antibody production was identified using the IEDB suite of antigenicity prediction software. The DNA sequence was amplified from RH genomic DNA and cloned into the pET-32 LIC/EK vector (Novagen), which adds an N-terminal thioredoxin and histidine tag to the expressed protein. Recombinant CP1Ag was expressed and initially purified via a nickel-affinity column (HisPur Thermo Fisher) as previously described (*Chasen et al., 2017*). Cleavage of the N-terminal thioredoxin and histidine tag was done by biotinylated thrombin. The antigen was passed again through the nickel column, and the purified tag-less antigen was gently eluted using 10 mM imidazole. Antibodies in mice were generated as previously (*Chasen et al., 2019*). Swiss Webster mice (Charles River) were inoculated intraperitoneally with 100 µg of TgTRPPL-2 mixed with complete Freund's adjuvant, followed by two boosts with 50 µg of TgTRPPL-2 in incomplete Freund's adjuvant. The final serum was collected by cardiac puncture after $CO_2$ euthanasia. We created a αSERCA antibody for co-localization studies of the TgTRPPL-2. The phosphorylation (P) and

nucleotide binding (N) domains of *Tg*SERCA were cloned into XmaI and HindIII sites of pQE-80L plasmid for expression in *Escherichia coli* BL21-CodonPlus competent cells. Purified antigen was used to immunize Guinea pigs with 0.2 mg of antigen mixed with Freund's complete adjuvant, followed by two boosts of 0.1 mg antigen mixed with Freund's incomplete adjuvant (Sigma F5506). The resulting antibodies were used at 1:1000 for western blots. The animal protocol used was approved by the UGA Institutional Animal Care and Use Committee (IACUC).

## Western blot analysis

SDS-polyacrylamide gel electrophoresis (SDS-PAGE) followed established protocols (*Laemmli, 1970*). Lysates were prepared by resuspending a pellet of $1 \times 10^8$ tachyzoites in 50 µl of Cell Lytic lysis buffer containing 12.5 U benzonase and 1X protease cocktail inhibitor (P8340 Sigma). The reaction was stopped with one volume of 2% SDS and 1 mM EDTA. Total lysates were boiled in Laemmli sample buffer (BioRad). Immunoblotting followed established protocols using mouse anti-HA monoclonal antibody (1:1000) (Roche). Detection was done using the Odyssey Clx LICOR system using goat anti-mouse IRDye800WC (1:10,000). Loading control for western blots was done with primary mouse-anti-tubulin antibodies at a 1:15,000 dilution and goat anti-mouse IRDDye800WC as secondary (1:10,000).

## Immunofluorescence microscopy

Extracellular parasites were collected and purified as previously (*Liu et al., 2014*). Parasites were washed once with buffer A with glucose (BAG, 116 mM NaCl, 5.4 mM KCl, 0.8 mM MgSO$_4$, 5.5 mM glucose, and 50 mM HEPES, pH 7.4), and an aliquot of $2 \times 10^4$ parasites was overlaid on a coverslip previously treated with poly-L-Lysine. Intracellular tachyzoites were grown on hTERT cells on coverslips. Both extracellular and intracellular parasites were fixed with 3% paraformaldehyde for 20 min at room temperature (RT), permeabilized with 0.3% Triton X-100, blocked with 3% bovine serum albumin (BSA), and exposed to primary antibodies (Ratα-HA 1:100). The secondary antibodies used were goat-αrat Alexa Fluor 488 (Life Technologies) at a 1:1000 dilution. For co-localization studies, we used α-Sag1 (1:1000) as membrane marker and α-TgSERCA as ER marker (1:1000). Slides were examined using an Olympus IX-71 inverted fluorescence microscope with a photometric CoolSNAP HQ charge-coupled device (CCD) camera driven by DeltaVision software (Applied Precision, Seattle, WA). Super-resolution images were imaged with a 63× oil (NA 1.4) objective on an 880-laser scanning microscope with Airyscan (Zeiss, Germany) with a 2× zoom. Airyscan images were process with the Zen Black Software (Zeiss, Germany).

## Immunoprecipitation assays

Freshly lysed tachyzoites expressing TgTRPPL-2-smHA were collected and filtered through an 8 µM membrane (Whatman). Tachyzoites were washed twice in BAG and resuspended in lysis buffer (50 mM Tris-HCl, pH 7.4, 150 mM KCl, 1 mM EDTA, 0.4% NP-40) to a final concentration of $2 \times 10^9$ total cells. Lysis was allowed to proceed for 30 min at 4°C, and cells were centrifuged at 15,000 × g for 20 min. Immunoprecipitation of TgTRPPL-2-smHA protein was performed using the Pierce HA Tag/Co-IP Kit (Thermo Fisher Scientific, Waltham, MA) according to the manufacturer's instructions. Briefly, HA magnetic beads were washed twice in lysis buffer and mixed with the parasite lysate by vortexing for 1 hr at RT. Beads were collected and the flow-through fraction was saved for further analysis. Beads were washed twice in wash buffer (50 mM Tris-HCl, pH 7.4, 150 mM KCl, 1 mM EDTA, 0.1% NP-40) and once in ddH$_2$O by gentle mixing. The tagged protein was recovered by mixing the beads with 1× Laemmli buffer and heated at 65°C for 10 min. The supernatant was collected and used for PAGE and western blots. The corresponding band was cut and resuspended in water and analyzed using LC-Mass Spectrometry. Samples were sent to the Proteomics and Mass Spectrometry Core Facility at the University of Georgia for analysis. The average counts that were obtained from two biological samples are shown in *Supplementary file 2*. Proteins with counts higher than three are shown.

## Growth and invasion assays

Plaque assays were done as previously described, with slight modifications (*Liu et al., 2014*). Briefly, 200 egressed tachyzoites were allowed to infect confluent hTERT cells for 7 days. After 7 days, cells

were fixed with ethanol and stained with crystal violet. Plaque sizes were analyzed using FIJI (*Schindelin et al., 2012*). Invasion assays were performed as previously described, with slight modifications (*Kafsack et al., 2004*). A subconfluent monolayer of HFF cells was infected with $2 \times 10^7$ tachyzoites in the presence of 1.8 mM or 0.5 mM $Ca^{2+}$ and placed for 20 min on ice and subsequently transferred for 5 min to a 37°C water bath for parasite invasion. Cells were immediately fixed with 3% paraformaldehyde for 20 min. Extracellular parasites (attached) were stained using RabbitαSag1 (1:1000) prior to permeabilization while intracellular parasites (invaded) were stained with MouseαSag1 (1:200). Secondary antibodies were goat-αrabbit Alexa Fluor 546 (1:1000) and goat-α mouse Alexa Fluor 488 (1:1000). Images were taken with an Olympus IX-71 inverted fluorescence microscope with a Photometric CoolSNAP HQ CCD camera driven by DeltaVision software (Applied Precision). Quantification was made of 10 fields of view at a 1000 magnification from three independent biological replicates. Percentage of invaded versus attached was quantified by dividing the number of parasites invaded or attached by the total parasites quantified in the field of view.

## Egress experiments

hTERT cells were infected with $5 \times 10^5$ of RH or ΔTgTRPPL-2 tachyzoites. 24 hr after infection, parasitophorous vacuoles (PVs) were observed by microscopy and washed with Ringer's buffer (155 mM NaCl, 3 mM KCl, 1 mM $MgCl_2$, 3 mM $NaH_2PO_4H_2O$, 10 mM HEPES, pH 7.3, and 5 mM glucose). Ringer's buffer was used as extracellular buffer in the presence or absence of 1.8 mM $Ca^{2+}$. Drugs were added in Ringer's buffer 30 s after imaging at the following concentrations: saponin (0.02%) or Zaprinast (100 μM). Images were acquired in a time-lapse mode with an acquisition rate of 3 s for 12–20 min. For statistical analysis, egress time was quantified as the first parasite to egress out of the PV. Statistical analysis was done for three independent biological replicates and at least 5 PVs per experiment.

## Cytosolic $Ca^{2+}$ measurements

Parasites were loaded with Fura-2-AM as described in *Stasic et al., 2021*; *Vella et al., 2019*. Briefly, fresh lysed extracellular tachyzoites were washed twice at 1800 rpm for 10 min at room temperature in buffer A (BAG) (116 mM NaCl, 5.4 mM KCl, 0.8 mM $MgSO_4$, 5.5 mM d-glucose, and 50 mM HEPES, pH 7.4). Parasites were resuspended to a final density of $1 \times 10^9$ parasites/ml in loading buffer (Ringer's plus 1.5% sucrose and 5 μM Fura-2-AM). The suspension was incubated for 26 min at 26°C with mild agitation. Subsequently, the parasites were washed twice with Ringer's buffer to remove extracellular dye. Parasites were resuspended to a final density of $1 \times 10^9$ parasites/ml in Ringer's buffer and kept in ice. For fluorescence measurements, $2 \times 10^7$ parasites/ml were placed in a cuvette with 2.5 ml of Ringer's buffer. The cuvette was placed in a thermostatically controlled Hitachi F-7000 fluorescence spectrophotometer. Excitation was at 340 and 380 nm, and emission at 510 nm. The Fura-2-AM fluorescence relationship to intracellular $Ca^{2+}$ concentration ($[Ca^{2+}]_i$) was calibrated from the ratio of 340/380 nm fluorescence values after subtraction of the background fluorescence of the cells at 340 and 380 nm as previously described (*Grynkiewicz et al., 1985*). Changes in $[Ca^{2+}]_i$ ($\Delta F [Ca^{2+}]$) were measured by subtracting the highest peak of $Ca^{2+}$ in the first 20 s after addition of $Ca^{2+}$ or 100 s after the addition of drugs minus the baseline.

## Cell transfections and culture of HEK-3KO cells

Total RNA of wild-type *T. gondii* were extracted and reversed transcribed into cDNA. TgTRPPL-2 whole cDNA was amplified using primers shown in *Supplementary file 3* (primers C1–C6). The amplified cDNA was cloned into the Zero Blunt TOPO vector using the cloning kit as per the manufacturer's instruction. Correct insertion was verified by colony PCR using M13F and M13R primers. Restriction digests was performed to remove the insert from the vector using the following restriction enzymes: BamHI and AvrII. The purified *TgTRPPL-2 cDNA* was ligated to linearized pCDNA 3.1 plasmid. Ligation to the vector was confirmed by PCR and sequencing. Purified TRPPL-2-pCDNA was used to co-transfect HEK-3KO cells.

Human embryonic kidney (HEK) cells, which have the three endogenous isoforms of the $IP_3$ receptor knocked out, were a gift from Dr. David Yule (*Alzayady et al., 2016*; *Lock et al., 2018*). The cells were maintained in DMEM with 10% fetal bovine serum 2.5 μg/ml amphotericin B and 100 μg/ml streptomycin. They are periodically checked for *Mycoplasma* contamination. Cells were transiently

transfected as previously described (*Longo et al., 2013*) with 2.5 µg of TgTRPPL-2, PC2, or RFP DNA targeted to the ER. Each plasmid DNA were diluted in 200 µl of Opti-MEM with 25 µl of polyethylenimine and incubated for 10 min. The mix was then added to semi-confluent HEK-3KO cells in a dropwise manner, and 24 hr later the media was changed.

## Preparation of nuclear extracts

48 hr after transfection, cells were collected and the nucleus extracted as previously described (*Mak et al., 2013a*). $2 \times 10^7$ of transiently transfected cells were collected in ice-cold PBS. Cells were spun down and washed twice in PBS and resuspended in Nuclei Isolation Solution (150 mM KCl, 250 mM sucrose, 10 mM Tris-HCl, 1.4 mM β-mercaptoethanol, 0.2 mM PMSF, pH 7.3). Cells were dounce homogenized and nuclei extracts stored on ice. 100 µl of nuclei were transferred to cover slips previously coated with poly-L-lysine and incubated for 20 min before filling the chamber with bath solution.

## Patch clamp of nuclear membranes

Nuclear extracts expressing TgTRPPL-2 or the control with gCaMPer (*Henderson et al., 2015*) were used for analysis. Currents were recorded using electrodes pulled from filamented borosilicate capillary glass (Harvard Bioscience, MA) with a resistance of 10–15 MΩ. After forming the seal and pulling the outside (cytosolic-side) out, membrane patch configurations (*Mak et al., 2013b*) holding potentials were maintained at 0 mV. Recordings were obtained using the HEKA Electronic Patch Clamp EPC10 (Harvard Bioscience). The internal solution contained (mM) 140 KCl or CsCl, 10 HEPES, 1.8 or 10 free $Ca^{2+}$ adjusted with EGTA. The Standard Bath Solution was symmetrical to the pipette solution except there was 100 nM or 10 µM of free $Ca^{2+}$. Currents were elicited by applying voltage steps from the holding potential of 0 mV and stepping down to −80 mV and then up to 20 mV over 25 s. Data was acquired at 45 kHz and filtered at 2 kHz. Each voltage sweep was conducted a total of five times. Analysis of current amplitude, channel open probability, and channel conductance was conducted using FitMaster (Harvard Bioscience). Conductance was calculated by plotting the current-voltage relationship for each condition tested and then determining the slope conductance between −80 to +20 mV. gCaMPer fluorescent signals were simultaneously analyzed as the current signals in preparations where the ER was broken into. A Lumencor LED light source switching between 488 and 561 nm was used to excite the samples, and the fluorescent signals captured on a Hamamatsu Flash 4.0 cMOS camera using Zeiss Zen Black software.

## Statistics

Statistical analyses were performed by Student's *t-test* using GraphPad PRISM version 8.2. All experimental data were analyzed from at least three independent biological replicates. Error bars shown represent standard error of the mean (SEM) of the biological replicates analyzed. For the electrophysiological analysis, a total of three cells per biological replicate (nine total cells) were analyzed.

## Acknowledgements

 We thank Drs. David Yule for providing the HEK-3KO cells for electrophysiological analyses, John Boothroyd for antibodies against SAG1, and Boris Striepen for providing us with the Cosmid for complementation. We thank Catherine Li for generating the antigen used to generate the anti-SERCA antibody. The super-resolution microscope is part of the Biomedical Microscopy Core (BMC) of the University of Georgia, which is directed by Dr. MK Kandasamy. We thank the University of Georgia Graduate School for awarding a Summer Research Travel Grant to KMMN. This work was supported by the U.S. National Institutes of Health grants AI154931 and AI128356 to SNJM and R00 DK101585 to IYK. KMMN was partially supported through a fellowship funded by a T32 training grant, 5T32AI060546.

## Additional information

### Funding

| Funder | Grant reference number | Author |
|---|---|---|
| National Institutes of Health | AI154931 | Silvia NJ Moreno |
| National Institutes of Health | AI128356 | Silvia NJ Moreno |
| National Institutes of Health | T32AI060546 | Karla Marie Márquez-Nogueras |
| National Institutes of Health | DK101585 | Ivana Y Kuo |

The funders had no role in study design, data collection and interpretation, or the decision to submit the work for publication.

### Author contributions

Karla Marie Márquez-Nogueras, Conceptualization, Validation, Investigation, Methodology, Writing - original draft, Writing - review and editing; Miryam Andrea Hortua Triana, Formal analysis, Investigation, Methodology; Nathan M Chasen, Resources, Investigation; Ivana Y Kuo, Conceptualization, Resources, Supervision, Funding acquisition, Methodology, Writing - review and editing; Silvia NJ Moreno, Conceptualization, Resources, Supervision, Funding acquisition, Project administration, Writing - review and editing

### Author ORCIDs

Karla Marie Márquez-Nogueras  https://orcid.org/0000-0002-9459-6882
Ivana Y Kuo  https://orcid.org/0000-0002-9867-2408
Silvia NJ Moreno  https://orcid.org/0000-0002-2041-6295

### Decision letter and Author response

Decision letter https://doi.org/10.7554/eLife.63417.sa1
Author response https://doi.org/10.7554/eLife.63417.sa2

## Additional files

### Supplementary files

• Source data 1. Original blots.

• Supplementary file 1. Top 10 hits resulting from the analysis of the TgGT1_310560 gene with the HHPRED server.

• Supplementary file 2. Mass spectrometry results of the sliced band after immunoprecipitation of TgTRPPL-2.

• Supplementary file 3. Primers used in this work.

• Supplementary file 4. Composition of the solutions used for the electrophysiological analysis.

• Transparent reporting form

### Data availability

All data generated or analysed during this study are included in the manuscript and supporting files.

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
