## [Decision Letter]

**Acceptance summary:**

This work identifies a Transient Receptor Potential (TRP) channel that regulates calcium signalling in the parasitic protist, *Toxoplasma gondii*. The TgTRPPL-2 protein localizes to both the plasma membrane and endoplasmic reticulum and has important roles in regulating influx of extracellular calcium, the release of intracellular calcium stores and key steps in parasite invasion and egress from host cells. The parasite channel is sensitive to broad spectrum TRP channel inhibitors suggesting that TgTRRPL-2 is a druggable target.

**Decision letter after peer review:**

Thank you for submitting your article "Calcium signaling through a Transient Receptor Channel is important for *Toxoplasma gondii* growth" for consideration by *eLife*. Your article has been reviewed by 3 peer reviewers, including Malcolm J McConville as the Reviewing Editor and Reviewer #1, and the evaluation has been overseen by Dominique Soldati-Favre as the Senior Editor. The following individuals involved in review of your submission have agreed to reveal their identity: Andrew Thomas (Reviewer #2) and Mathieu Brochet and Aurélia Balestra (Reviewer #3).

The reviewers have discussed the reviews with one another and the Reviewing Editor has drafted this decision to help you prepare a revised submission.

Summary:

This paper provides strong evidence that a putative *Toxoplasma gondii* TRP cation channel protein, TgTRPPL-2, mediates calcium influx at the plasma membrane and calcium efflux in the ER. The dual localization of both epitope-tagged and native TgTRPPL protein to the PM and ER was assessed by microscopy, while phenotypic analysis of a ∆TgTRPPL-2 parasite line indicated that the channel is required for parasite invasion and egress in vitro. Calcium transport activity was measured using patch-clamp techniques in HEK3KO cells expressing TfTRPPL-2 as well as the use of well-defined inhibitors. The findings add to our current understanding of calcium signalling in these evolutionarily divergent eukaryotes, which has been shown by many studies to underpin many signalling and differentiation events in the *T. gondii* life cycle.

Essential revisions:

1. The localization of epitope-tagged and native TgTRPPL-2 to the ER is compelling. However, the evidence for plasma membrane localization is not as clear and should be supported by additional evidence (including higher resolution images) and/or more careful discussion. In particular, can the authors discount the possibility that TgTRPPL-2 also localizes to the IMC, which has the potential to be a calcium store.

2. The authors need to further clarify their gene/protein model and better justify their proposal that TgTRPPL is cleaved into two proteins/protein isoforms in vivo. Specifically, do the authors have any evidence of different sized proteins in parasite lines expressing epitope tagged proteins or from using antibodies to the native protein(s)?

3. The authors should clarify the reverse genetic strategy and the experimental conditions used to phenotype the parasites. In particular, while Figure 2A and the text suggest a disruption of trppl2, the method indicates a full deletion. A clear figure depicting the strategy and showing genotyping should be included.

4. The authors propose that nifedipine increases basal cytosolic Ca^2+^ selectively in the in ΔTgTRPPL-2 mutant (Figure 6B) by increasing leakage from intracellular Ca^2+^ stores as a means to compensate for the absence of plasma membrane Ca^2+^ influx. However, it is unclear how the absence of plasma membrane Ca^2+^ entry could couple to enhanced leak from intracellular stores. Moreover, there is no mechanistic explanation for why sensitivity to nifedipine is lost in ΔTgTRPPL-2 and then restored in the complemented mutant (Figure 6D). The authors should further qualify this hypothesis or remove it.

---

## [Author Response]

Essential revisions:1. The localization of epitope-tagged and native TgTRPPL-2 to the ER is compelling. However, the evidence for plasma membrane localization is not as clear and should be supported by additional evidence (including higher resolution images) and/or more careful discussion. In particular, can the authors discount the possibility that TgTRPPL-2 also localizes to the IMC, which has the potential to be a calcium store.

We now include additional IFA experiments of colocalization with the SAG surface marker using super resolution microscopy and Gap45, which is anchored to the Inner Membrane complex. We present the data in Figure 1—figure supplement 1 and panel D of Figure 1. We have modified the text to include the information in the Results section, lines 131-137 and 144-149. According to this result TgTRPPL-2 colocalizes better with the PM marker SAG1, which is supported with the images at ~140 nm resolution as shown in Figure 1—figure supplement 1. The expression level of TgTRPP2 is low but its localization to the ER is very clear. The peripheral localization appears as vesicles in contact with the plasma membrane.

2. The authors need to further clarify their gene/protein model and better justify their proposal that TgTRPPL is cleaved into two proteins/protein isoforms in vivo. Specifically, do the authors have any evidence of different sized proteins in parasite lines expressing epitope tagged proteins or from using antibodies to the native protein(s)?

We now include a western blot (Figure 2—figure supplement 1B) using 4 different cell lines (1. TatiDKu80, 2. TgTRPPL2-smHA, 3. RH and 4. DTgTRRPL2) with the native anti-TgTRPPL2 antibody. In all the cell lines there is a band at ~115-120 kDa, which represents the endogenous cleaved TgTRPPL2. In TgTRPPL2-smHA tachyzoites there is an additional band of ~150 kDa due to the smHA tag which adds approximately 39 kDa to the protein. The sum of these two bands (120 + 150kDa) represents the size predicted for TgTRPPL2 tagged with the smHA tag in ToxoDB. This is discussed in lines 165-173.

3. The authors should clarify the reverse genetic strategy and the experimental conditions used to phenotype the parasites. In particular, while Figure 2A and the text suggest a disruption of trppl2, the method indicates a full deletion. A clear figure depicting the strategy and showing genotyping should be included.

We have included an electrophoresis gel showing genotyping of the insertion (Figure 2—figure supplement 1A). Although the method is designed for deletion, we used the RH strain, which uses Non-homologous end joining repair and homologous recombination is not favored. In this case CRISPR KOs resulted in the insertion of the drug cassette disrupting the transcription of TgTRPPL2. We have validated that the insertion did disrupt the transcription and translation of the TgTRPPL-2 gene by generating an antibody that confirms the absence of TgTRPPL2 in the mutant by both IFA (Figure 2—figure supplement 1 C and D) and western blots (Figure 2—figure supplement 1B), in addition to quantification of RNA transcripts by qPCR (Figure 2B).

4. The authors propose that nifedipine increases basal cytosolic Ca^2+^ selectively in the in ΔTgTRPPL-2 mutant (Figure 6B) by increasing leakage from intracellular Ca^2+^ stores as a means to compensate for the absence of plasma membrane Ca^2+^ influx. However, it is unclear how the absence of plasma membrane Ca^2+^ entry could couple to enhanced leak from intracellular stores. Moreover, there is no mechanistic explanation for why sensitivity to nifedipine is lost in ΔTgTRPPL-2 and then restored in the complemented mutant (Figure 6D). The authors should further qualify this hypothesis or remove it.

We have consistently found that incubation with nifedipine increases cytosolic calcium in the mutant cell line and unfortunately, we don’t completely understand this phenomenon. We are not sure if this is related to the activity of TgTRPPL-2 at the PM or the ER. It is possible that nifedipine has a secondary effect on a Ca^2+^ pump that pumps Ca^2+^ from the cytosol. It could be that the pump interacts with TgTRPPL2 and because of the missing protein the pump is more sensitive to inhibition. This of course is highly speculative, and it would require further investigation. For now, we deleted the figure and redefined Calcium influx as the DCa^2+^ following the addition of extracellular Ca^2+^ and show the quantification (Figure 6D). We do not believe that the ΔTgTRPPL-2 mutant lost sensitivity to nifedipine. We think that they are more sensitive (see quantification in 6D, under D). We think that it is hard to separate the activity of TgTRPPL-2 from that of the other channels at the membrane because of the regulatory function of Ca^2+^ itself on Ca^2+^ transport. We have re-arranged the figure and explained the results in the text, lines 342-347.